

# Simulating shrubs and their energy and carbon dioxide fluxes in Canada's Low Arctic with the Canadian Land Surface Scheme Including biogeochemical Cycles (CLASSIC)

Gesa Meyer[1,2], Elyn R. Humphreys[2], Joe R. Melton[1], Alex J. Cannon[1], and Peter M. Lafleur[3]

[1]Environment and Climate Change Canada, Climate Research Division, Victoria, BC, Canada
[2]Carleton University, Geography and Environmental Studies, Ottawa, ON, Canada
[3]Trent University, School of Environment, Peterborough, ON, Canada

**Correspondence:** Gesa Meyer (gesa.meyer@canada.ca)

**Abstract.** The Arctic is warming more rapidly than other regions of the world leading to ecosystem change including shifts in vegetation communities, permafrost degradation and alteration of tundra surface-atmosphere energy and carbon (C) fluxes, among others. However, year-round C and energy flux measurements at high-latitude sites remain rare. This poses a challenge for evaluating the impacts of climate change on Arctic tundra ecosystems and for developing and evaluating process-based

models, which may be used to predict regional and global energy and C feedbacks to the climate system. Our study used 14 years of seasonal eddy covariance (EC) measurements of carbon dioxide ($CO_2$), water and energy fluxes and winter soil chamber $CO_2$ flux measurements at a dwarf-shrub tundra site underlain by continuous permafrost in Canada's Southern Arctic ecozone to evaluate the incorporation of shrub plant functional types (PFTs) in the Canadian Land Surface Scheme Including biogeochemical Cycles (CLASSIC), the land surface component of the Canadian Earth System Model. In addition to new

PFTs, a modification of the efficiency with which water evaporates from the ground surface was applied. This modification addressed a high ground evaporation bias that reduced model performance when soils became very dry, limited heat flow into the ground and reduced plant productivity through water stress effects. Compared to the grass and tree PFTs previously used by CLASSIC to represent the vegetation in Arctic permafrost-affected regions, simulations with the new shrub PFTs better capture the physical and biogeochemical impact of shrubs on the magnitude and seasonality of energy and $CO_2$ fluxes at the

dwarf-shrub tundra evaluation site. The revised model, however, tends to overestimate gross primary productivity, particularly in spring, and overestimated late winter $CO_2$ emissions. On average, annual net ecosystem $CO_2$ exchange was positive for all simulations, suggesting this site was a net $CO_2$ source of $18 \pm 4$ g C m$^{-2}$ year$^{-1}$ using shrub PFTs, $15 \pm 6$ g C m$^{-2}$ year$^{-1}$ using grass PFTs, and $25 \pm 5$ g C m$^{-2}$ year$^{-1}$ using tree PFTs. These results highlight the importance of using appropriate PFTs in process-based models to simulate current and future Arctic surface-atmosphere interactions.



## 1 Introduction

The terrestrial carbon (C) cycle of the Arctic is changing as the region warms at more than twice the rate of the rest of the
world (IPCC, 2014; Post et al., 2019). Enhanced Arctic soil C loss to the atmosphere and waterways is linked to permafrost
degradation and thermokarst processes, deeper active layers, deeper snow, and more frequent and intense fires (Hayes et al.,
2014; Abbott et al., 2016; Myers-Smith et al., 2019; Myers-Smith et al., 2020; Belshe et al., 2013; Turetsky et al., 2020). A
significant positive climate feedback effect is possible if even a small proportion of the approximately $1035 \pm 150$ Pg C stored
in the top 3 m of Arctic soils (Hugelius et al., 2014) is transferred to the atmosphere (Chapin et al., 2005; Schuur et al., 2009;
Hayes et al., 2014). However, longer growing seasons and Arctic 'greening' associated, in part, with northward migration of
the tree line and shrub expansion are linked to increased growing season carbon dioxide ($CO_2$) uptake (Belshe et al., 2013;
Abbott et al., 2016; Myers-Smith et al., 2011).

There are large uncertainties as to whether the Arctic tundra is currently an annual source or sink of $CO_2$ (McGuire et al.,
2012). Recent studies have highlighted the importance of $CO_2$ emissions during the long winter and shoulder season periods,
which may offset $CO_2$ gains by Arctic ecosystems during the short growing season (Belshe et al., 2013; Oechel et al., 2014;
Euskirchen et al., 2017; Arndt et al., 2020). Long-term trends in winter $CO_2$ fluxes are generally difficult to ascertain due to
a scarcity of year-round measurements, but several studies have suggested winter $CO_2$ emissions are changing. Belshe et al.
(2013) found a significant increase in winter $CO_2$ emissions over the 2004-2010 time period using observations from six pan-
Arctic sites. Natali et al. (2019) up-scaled in-situ measurements from about 100 high-latitude sites using a boosted regression
tree machine learning model to estimate a contemporary loss of 1,662 Tg C per year from the Arctic and boreal northern
permafrost region (land area $\geq 49°$N) from October to April (2003-2017). This winter loss of $CO_2$ exceeded the average
growing season $CO_2$ uptake estimated using five process-based models by over 600 Tg C per year. Using hourly atmospheric
$CO_2$ measurements, early winter respiration rates from Alaska's North Slope tundra region are estimated to have increased by
$73 \pm 11\%$ since 1975 (Commane et al., 2017). However, both growing season and winter $CO_2$ fluxes from Alaska's North
Slope were not well represented by the majority of 11 Earth System Models from the Coupled Model Intercomparison Project
Phase 5 (CMIP5) investigated by Commane et al. (2017). Most models predicted the start of the growing season earlier than
observed, underestimated early winter $CO_2$ losses and overestimated annual net $CO_2$ uptake (Commane et al., 2017).

Field observations and process-based models are complementary approaches to better understand Arctic $CO_2$ sink and source
dynamics. Field observations may be used to parameterize and evaluate process-based models (Zhang et al., 2014; Commane
et al., 2017; Park et al., 2018; Huntzinger et al., 2020) which are then applied over larger regions (McGuire et al., 2012; Jeong
et al., 2018; Ciais et al., 2019) and used to project C flux trends into the future (McGuire et al., 2018; Post et al., 2019). Model
estimates of C budgets are especially important in the Arctic, as year-round flux measurements are difficult to obtain and hence
remain rare.

It is critical that land surface models are able to capture the important interactions between the Arctic tundra and the atmo-
sphere due to potential feedbacks on regional and global climate. The Canadian Land Surface Scheme Including Biogeochem-
ical Cycles (CLASSIC, the open-source community model successor to CLASS-CTEM; Melton et al., 2020) represents the





land surface exchanges of energy, water, and C in the Canadian Earth System Model (CanESM) (Swart et al., 2019) and has been extensively evaluated on a global-scale (e.g. Melton and Arora, 2016; Arora et al., 2018). CLASSIC already focuses on several physical processes relevant to the high latitudes including treatment of snow and soil freeze/thaw processes (Melton

et al., 2019) and peatland C cycling (Wu et al., 2016). However, CLASSIC does not have a plant functional type (PFT) specific to tundra, instead tundra vegetation is represented by $C_3$ grass and/or trees depending on land cover mapping. In reality, Arctic vegetation is diverse consisting of erect or prostrate, evergreen or deciduous shrubs, graminoids, herbs, moss, and lichen (Chapin and Shaver, 1985). Although there are challenges in interpreting satellite-based trends of Arctic greening or browning (Myers-Smith et al., 2020), observed increases in greenness or productivity have been linked to shrub expansion through infill-

ing of patches, advances in shrubline, and increased height and abundance of shrub species (Myers-Smith et al., 2011; Lantz et al., 2012). These kind of changes in tundra vegetation communities can affect snow and active layer depths, hydrology, surface-atmosphere energy exchange, nutrient dynamics and the terrestrial C balance of the Arctic tundra (Myers-Smith et al., 2011).

In order to further improve the representation of Arctic surface-atmosphere interactions in CLASSIC, we evaluate new

dwarf deciduous and evergreen shrubs and sedge PFT parameterizations with EC-based observations of $CO_2$ and energy fluxes at an erect dwarf-shrub tundra site in Canada's Southern Arctic. We also address a high soil evaporation bias discovered in CLASSIC, which has important implications for appropriately simulating energy flux feedbacks and water-stress impacts on growing season photosynthesis. We compare our off-line model simulations with $C_3$ grass and tree parameterizations to highlight how shrubs uniquely affect $CO_2$ and energy exchange with the atmosphere. Finally, we use CLASSIC to simulate

winter $CO_2$ fluxes and estimate the annual $CO_2$ budget for this tundra site where winter flux measurements are challenging to obtain.

## 2 Methods

### 2.1 CLASSIC

#### 2.1.1 Model description

A detailed description of CLASSIC v1.0 can be found in Melton et al. (2020). It couples a land surface physics sub-module (the Canadian Land Surface Scheme; CLASS; Verseghy, 2017) to a biogeochemistry sub-module (the Canadian Terrestrial Ecosystem Model; CTEM; Melton and Arora, 2016) as an open-source community successor to CLASS-CTEM. Physical processes determining energy and water balances of the land surface are implemented in CLASS, described in detail in Verseghy (1991, 2000) and Verseghy et al. (1993), and biogeochemical processes are handled by CTEM, described in detail in Arora

(2003) and Melton and Arora (2016).

The atmospheric forcing variables required by CLASSIC are incoming shortwave ($R_{SW}$) and longwave radiation ($R_{LW}$), air temperature ($T_a$), the precipitation rate, air pressure (PA), specific humidity ($q$) and wind speed ($U$). Driven by these meteorological data, the transfer of heat and water through the soil layers and snow, as well as the exchange with the atmosphere





and within the vegetation canopy, are typically calculated on a half-hourly time step when run offline. Net radiation ($R_n$) is
calculated using prognostically determined ground and snow albedo, the land surface temperature, $R_{SW}$ and $R_{LW}$. CLASSIC
enforces energy balance closure. In previous versions of CLASS, soil layers were limited to 3 layers with a maximum soil
depth of 4.1 m, but CLASSIC allows for an arbitrary number of ground layers and deeper depths. In this analysis, we used
22 layers starting with a thickness of 10 cm, which increased with depth, down to a depth of 20 m (Supplementary Materials,
Table S1) following the recommendations by Melton et al. (2019) for permafrost-affected soils. Water movement between the
95 soil layers occurs until the bottom of the permeable layer (set to 5 m in this study), while heat transfer is taken into account
within the whole ground column thus including the permeable layers and the bedrock below. Energy, momentum and water
fluxes are calculated on a 30 minute time step including photosynthesis and canopy conductance. These fluxes are influenced
by vegetation characteristics such as vegetation height, leaf area index (LAI), leaf and stem biomass and rooting depth. These
vegetation characteristics depend on PFT parameterizations and are dynamically determined within the biogeochemistry sub-
100 module through the allocation of C on a daily time step, which use the accumulated photosynthetic fluxes calculated on the
physics time step. In addition to C allocation to leaves, stems and roots, the biogeochemistry sub-module simulates other
biogeochemical processes such as autotrophic ($R_a$) and heterotrophic respiration ($R_h$) from its leaf, stem, root, litter and soil
C pools on a daily time step. The biogeochemistry sub-module obtains required information about the land surface, e.g., $R_n$ ,
soil temperatures and water content (liquid and frozen), from the physics sub-module.

**2.1.2   Model modifications**

CLASSIC v1.0.1 uses 4 broad categories of PFTs (needleleaf trees, broadleaf trees, crops and grasses) in its calculation of
physical land surface processes (relating to energy, momentum and water) in the physics sub-module. These are expanded to
9 PFTs for simulating biogeochemical processes, to account for deciduous or evergreen habits (e.g., in their phenology) or for
plants with $C_3$ versus $C_4$ photosynthetic pathways.

In this study, we added one more broad category PFT to the physics sub-module (shrubs) and three more PFTs to the
biogeochemistry sub-module (cold broadleaf deciduous shrubs, broadleaf evergreen shrubs, and sedges). The biogeochemistry
PFTs map onto the physics PFTs as shown in Table 1. Sedges are considered a grass by the physics sub-module while evergreen
and cold deciduous shrubs are assigned to the shrub PFT. The parameterizations developed by Wu et al. (2016) for the two
shrub PFTs and a sedge PFT for a peatland-specific sub-module for CLASS-CTEM were used as a starting point. Note that
Wu et al. (2016)'s peatland sub-module PFTs only considered the biogeochemical impacts of these PFTs. We fully integrate
these new PFTs in order to simulate the physical impact of the shrub's unique growth form and function.

The original CLASSIC formulation of ground evaporation, $E$, tended to overestimate $E$ during periods of water ponding on
the ground surface or when the water content of the top soil layer, $\theta_1$ (m$^3$ m$^{-3}$), was high, e.g., during and after snowmelt (see
Section 3.3 and Figure 4c). This excessive $E$ dried the soil to such an extent that summer evapotranspiration (ET) and gross
primary productivity (GPP) were underestimated due to water stress. This issue was also observed at shrubland sites by Sun
and Verseghy (2019), who reduced soil $E$ by applying a site- or soil texture-specific scaling factor to the maximum surface
evaporation rate($E(0)_{max}$). For a more broadly applicable formulation, we adopted the approach of Merlin et al. (2011), which





**Table 1.** Mapping between plant functional types (PFTs) used in CLASSIC's physics and biogeochemical calculations. PFTs in bold font indicate the new PFTs incorporated by our study.

| PFTs used in model physics calculations | PFTs used in model biogeochemical calculations | | |
|---|---|---|---|
| Needleleaf Tree | Evergreen | Deciduous | |
| Broadleaf Tree | Evergreen | Cold Deciduous | Drought/Dry Deciduous |
| Crop | CropC3 | CropC4 | |
| Grass | GrassC3 | GrassC4 | **Sedge** |
| **Broadleaf Shrub** | **Evergreen** | **Deciduous** | |

modifies the parameterization of the evaporation efficiency coefficient ($\beta$). In CLASSIC, the potential evaporation rate from bare soil, $E(0)$ (mm s$^{-1}$), is calculated as

$$E(0) = \rho_a C_{DH} V_a (q(0) - q_a), \tag{1}$$

where $\rho_a$ is the air density (kg m$^{-3}$), $C_{DH}$ is the surface drag coefficient for evaporation (unitless), $V_a$ is the wind speed at reference height (m s$^{-1}$), $q(0)$ is the specific humidity at the surface (kg kg$^{-1}$) and $q_a$ at the reference height (kg kg$^{-1}$) (Verseghy, 2017). The surface evaporation rate is capped by a maximum value, $E(0)_{max}$ determined by $\theta_1$ and the depth of water ponded on the surface ($Z_p$, m) as

$$E(0)_{max} = \rho_w (Z_p + (\theta_1 - \theta_{min})\Delta Z_1)/\Delta t, \tag{2}$$

with the water density $\rho_w$ (kg m$^{-3}$), the residual soil liquid water content remaining after freezing or evaporation $\theta_{min}$ (m$^3$ m$^{-3}$), which is set to 0.04 m$^3$ m$^{-3}$ for mineral and fibric organic soils, the depth of the top soil layer $\Delta Z_1$ (e.g. 0.10 m) and the time interval $\Delta t$ (s)(Verseghy, 2017). The saturated surface specific humidity, $q(0)_{sat}$, $q_a$ and $\beta$ determine $q(0)$ as

$$q(0) = \beta q(0)_{sat} + (1 - \beta)q_a, \tag{3}$$

where $\beta$ is defined using a relation by Lee and Pielke (1992) as

$$\beta = \begin{cases} 0 & \text{for } \theta_1 < \theta_{min} \\ 0.25(1 - \cos(\pi\theta_1/\theta_{fc}))^2 & \text{for } \theta_{min} < \theta_1 \leq \theta_{fc} \\ 1 & \text{for } \theta_1 > \theta_{fc}, \end{cases} \tag{4}$$

where $\theta_{fc}$ is the field capacity of the top soil layer (m$^3$ m$^{-3}$). Merlin et al. (2011) suggested using the volumetric water content at saturation as the maximum water content instead of $\theta_{fc}$ based on the fact that the quasi-instantaneous process of potential evaporation is physically reached at saturation. Thus, the definition of $\beta$ is modified to

$$\beta = \begin{cases} 0 & \text{for } \theta_1 < \theta_{min} \\ 0.25(1 - \cos(\pi\theta_1/\theta_p))^2 & \text{for } \theta_{min} < \theta_1 \leq \theta_p. \end{cases} \tag{5}$$





where $\theta_p$ is the porosity of the top soil layer (m$^3$ m$^{-3}$). Therefore, Equation 5 limits $\beta$ to values below 1 except when soils are fully saturated which differs from the previous parameterization (Equation 4), where $\beta$ remained 1 until $\theta$ was less than $\theta_{fc}$.

To incorporate shrub and sedge PFTs, several more modifications were made to CLASSIC. As the photosynthetic capacity of Arctic shrubs is seasonally variable and has been shown to depend on day length and maximum insolation (Chapin and

Shaver, 1985; Shaver and Kummerow, 1992; Oberbauer et al., 2013), especially in the fall, a seasonal variation of the maximum carboxylation rate by the Rubisco enzyme ($V_{cmax}$, mol CO$_2$ m$^{-2}$ s$^{-1}$) was implemented for shrubs as it was previously included for deciduous tree species in CLASSIC. Following Bauerle et al. (2012) and Alton (2017), seasonality was included by modifying $V_{cmax}$ such that

$$V_{cmax,new} = V_{cmax}(\frac{\mathrm{d}ayl}{\mathrm{d}ayl_{max}})^2 \tag{6}$$

with the current ($\mathrm{d}ayl$, hours) and annual maximum day length ($\mathrm{d}ayl_{max}$, hours) determined from the site's latitude. $V_{cmax}$ affects the maximum catalytic capacity of Rubisco ($V_m$, mol CO$_2$ m$^{-2}$ s$^{-1}$; Melton and Arora, 2016).

Vegetation height, $h$ (m), depends on stem biomass ($C_s$; kg C m$^{-2}$) in CLASSIC. Shrub $h$ is calculated following Wu et al. (2016) as $h = \min(0.25C_s^{0.2}, 4.0)$, but here we now allow for taller shrubs with a maximum height of 4 m instead of 1 m. Like grass, but unlike trees, shrubs can be buried by snow in CLASSIC.

Table 2 lists the parameters that were adapted for the new shrub PFTs (see Supplementary Materials for equations) including the leaf life span ($\tau_L$, years), specific leaf area (SLA, m$^2$ kg$^{-1}$ C), maximum vegetation age ($A_{max}$, years), and the parameter $\bar{\iota}$, which determines the root profile and thus affects rooting depth ($d_R$, m). The minimum root to shoot ratio ($lr_{min}$) for shrubs determines the allocation of C between roots and above-ground biomass in stems and leaves and was obtained from Qi et al. (2019).

In C allocation calculations, the C in stems ($C_s$, kg C m$^{-2}$), roots ($C_R$, kg C m$^{-2}$) and leaves ($C_L$, kg C m$^{-2}$) has to satisfy the following relationship

$$C_s + C_R = \eta C_L^{\kappa} \tag{7}$$

(Melton and Arora, 2016) while also meeting the $lr_{min}$ condition

$$\frac{C_R}{C_S + C_L} \geq lr_{min}. \tag{8}$$

If the root to shoot ratio falls below $lr_{min}$, C is preferentially allocated to roots (Melton and Arora, 2016). The parameters $\eta$ and $\kappa$ are PFT-specific (Table 2). The parameter values for trees, crops and grasses are based on Ludeke et al. (1994), while $\eta$ and $\kappa$ for shrubs were estimated from values of $C_s, C_R$ and $C_L$ for shrub tundra (Nobrega and Grogan, 2007; Grogan and Chapin, 2000; Murphy et al., 2009; Wang et al., 2016), resulting in the same $\kappa$ values as for grasses, but higher $\eta$ (Table 2).



**Table 2.** CLASSIC parameter values for the PFTs used in this study including the new PFTs (Sedge, Broadleaf Evergreen Shrub and Broadleaf Deciduous Cold Shrub). Equation numbers refer to those in this text or the Supplementary Materials.

| Parameter | Equation | Units | Sedge | Broadleaf Evergreen Shrub | Broadleaf Deciduous Cold Shrub | C3 grass | Needleleaf Evergreen Tree | Needleleaf Deciduous Tree |
|---|---|---|---|---|---|---|---|---|
| $V_{cmax}$ | 6 | $\mu$mol $CO_2$ m$^{-2}$ s$^{-1}$ | 40 | 60 | 60 | 55 | 42 | 47 |
| SLA | S2 | m$^2$ kg$^{-1}$ | 10 | 8 | 15 | -[a] | -[a] | -[a] |
| $\tau_L$ | S1 | years | 1 | 2 | 1 | 1 | 5 | 1 |
| $lr_{min}$ | 8 | dimensionless | 0.30 | 1.68 | 1.68 | 0.50 | 0.16 | 0.16 |
| $\bar{\iota}$ | S6 | dimensionless | 9.50 | 4.70 | 5.86 | 5.86 | 4.70 | 5.86 |
| $\eta$ | 7 | dimensionless | 3.0 | 6.0 | 6.0 | 3.0 | 10.0 | 30.8 |
| $\kappa$ | 7 | dimensionless | 1.2 | 1.2 | 1.2 | 1.2 | 1.6 | 1.6 |
| $A_{max}$ | S3 | years | N/A[b] | 100 | 100 | N/A[b] | 250 | 400 |

[a]Determined by the model from the leaf life span.

[b]In CLASSIC, $A_{max}$ is not defined for grasses and sedges, as the age related mortality is not applied to these PFTs.

## 170  2.2  Model evaluation dataset

### 2.2.1  Study site

The study site at Daring Lake (AmeriFlux designation CA-DL1, hereafter referred to as DL1, 64°52.131'N, 111°34.498'W) is located in Canada's Northwest Territories, approximately 300 km northeast of Yellowknife, at an elevation of 425 m. The climate is characterized by short summers and long, cold winters with a mean annual air temperature of -8.9°C (Lafleur and
Humphreys, 2018) and 200 to 300 mm of precipitation on average (ECG, 2012). In better drained areas, the surface soil organic layer is typically shallow, ranging between 1 to 10 cm in depth, deepening to 20 cm or more in wetter areas, with all areas underlain by coarse textured mineral soil (sand to loamy sand) (Humphreys and Lafleur, 2011). The average thaw depth in late summer is $86 \pm 3$ cm ($\pm$ SE) (Lafleur and Humphreys, 2018) measured over the period 2010-2015 using a metal probe inserted into the soil at 40 points (10 points every 5 m in the four cardinal directions around the measurement tower). DL1
is approximately 70 km north of the treeline in Canada's Southern Arctic ecozone (ECG, 2012). The dominant vegetation at DL1 includes evergreen shrubs (*Rhododendron tomentosum*, *Empetrum nigrum*, *Loiseleuria procumbens*) and deciduous shrubs (*Betula glandulosa*, *Vaccinum uliginosum*) (Lafleur and Humphreys, 2018). Small variations in microtopography result in wet areas covering about 10% of the area within 100 m of the measurement tower, which support tussock-forming sedges and *Sphagnum* species. Growing season vascular plant cover at DL1 was $63.6 \pm 5.4$ % determined using point frame measurements
in July 2019 at 10 quadrats with 25 points each. During the study period, the mean LAI during July measured using a plant canopy analyzer (model LAI-2200, LI-COR Inc., Lincoln, NE, USA, (LI-COR)) at 10 plots was $0.52 \pm 0.05$ ($\pm$ SE) m$^2$m$^{-2}$, mean height of shrubs was $18.2 \pm 1.3$ cm (Lafleur and Humphreys, 2018), and ground cover included mosses (percent ground cover: $16.5 \pm 4.9$ %) and lichens ($78.9 \pm 5.0$ %) (Lafleur and Humphreys, 2018).





### 2.2.2 Measurements and data processing

Eddy covariance measurements of turbulent $CO_2$ flux and energy fluxes, latent (LE) and sensible ($H$) heat flux, have been made at DL1 since 2004. The measurements and data processing are described in detail by Lafleur and Humphreys (2018). Briefly, the EC system consists of a $CO_2/H_2O$ infrared gas analyzer (IRGA) and a three-dimensional sonic anemometer (model R3-50, Gill Instruments, Lymington, UK) operating at 10 Hz. An open-path IRGA (model LI-7500, LI-COR) was operated between 2004 and 2015 while an enclosed-path sensor (LI-7200, LI-COR) was operated since 2014. The two IRGAs were run

concurrently for 2014 and 2015 to develop a site-specific correction for the self-heating issue with the LI-7500 (Burba et al., 2008) described by Lafleur and Humphreys (2018). This correction was applied to all of the LI-7500 data. Half-hourly fluxes are calculated using EddyPro$^{TM}$ (v. 6.2.0) (LI-COR) with block averaging, a double coordinate rotation, and no angle of attack correction. When the open-path IRGA operated, density fluctuations were addressed using the Webb et al. (1980) approach and when the enclosed-path IRGA operated, fluxes were computed directly from $CO_2/H_2O$ mixing ratios. The covariance of

the vertical wind speed and IRGA signals were used to compute time lags. Analytic correction of high- (Moncrieff et al., 2004) and low-pass (Moncrieff et al., 1997) filtering effects was applied. Half-hourly fluxes were removed from the time series due to sensor errors, power loss, or when associated variables (e.g. vertical velocity, $CO_2/H_2O$ concentrations, etc. were outside acceptable ranges). A 0.1 m s$^{-1}$ friction velocity threshold was applied to $CO_2$ fluxes at night and during snow-covered periods (Lafleur and Humphreys, 2008). Half-hourly net ecosystem productivity (NEP) was calculated as the sum of the $CO_2$

flux and the rate of change in $CO_2$ storage below the 4 m measurement height. The daily sum of turbulent energy fluxes did not equal available energy most days (e.g., mean daily $\frac{H+\text{LE}}{R_n - G}$ varied from 90-95% mid-summer to 67% on average during the snow-melt period (Figure S5); note that changes in energy storage between the EC instrumentation and the ground surface were not included in the evaluation of energy balance closure). LE and $H$ were not adjusted for energy balance closure.

Half-hourly NEP was partitioned into GPP and ecosystem respiration ($R_e$) using methods similar to those described by

Reichstein et al. (2005). An exponential temperature response function (Lloyd and Taylor, 1994) was parameterized using nighttime measurements of NEP (i.e., $R_e$) and $T_a$,

$$R_e = R_{ref} e^{E_0 \left( \frac{1}{T_{ref} - T_0} - \frac{1}{T_a - T_0} \right)} \tag{9}$$

with $T_{ref}$ set to 10°C and $T_0$ is -46.02°C. Equation 9 was first fit to the measurements within a moving window period of 15 days moved in increments of 5 days. The average of all temperature sensitivity ($E_0$) estimates which met the criteria (between 0

and 450 K) was calculated (144.7 K) and applied to Equation 9 to estimate the temperature independent respiration rate ($R_{ref}$) within consecutive 4-day periods. Finally, the constant $E_0$ and linearly interpolated $R_{ref}$ values were used with Equation 9 to calculate $R_e$ for all daytime half hours and nighttime half hours without measurements, as a means to gap-fill nighttime NEP. GPP was calculated as the sum of NEP and $R_e$. Missing daytime half-hourly estimates of GPP were gap-filled by fitting a light response curve to all growing season GPP and PAR (Equation 10) and adjusting the resulting GPP estimates by regressing

these against previously estimated values within consecutive 4-day periods.

$$\text{GPP} = \frac{GPP_{max} \alpha \text{PAR}}{\alpha \text{PAR} + GPP_{max}}, \tag{10}$$





where $GPP_{max}$ ($\mu$mol m$^{-2}$ s$^{-1}$) is the maximum photosynthetic capacity at light saturation and $\alpha$ (mol CO$_2$ (mol PAR)$^{-1}$) the quantum efficiency.

Cold season GPP was assumed to be negligible and thus NEP was equal to $R_e$ starting in the fall when the average $T_a$

remained below -1°C for 3 consecutive days after September 1$^{st}$ of each year and until the snow had melted. Cold season NEP and $R_e$ were gap-filled using the average cold season observations during the 4-day window period as Equation 9 was often poorly fitted during these periods. CO$_2$ is removed from the atmosphere and taken up by the ecosystem when NEP and GPP values are positive. Positive $R_e$ indicate an emission of CO$_2$ from the ecosystem to the atmosphere.

Latent heat flux was gap-filled using daytime and nighttime regressions between LE and available energy for all summer

measurements. Estimates of LE were adjusted using a multiplier to match observed LE within 4-day consecutive periods. Sensible heat flux was gap-filled as the difference between available energy and gap-filled LE adjusted by a multiplier to account for changing energy budget imbalance. Daily total CO$_2$ and energy fluxes were calculated from gap-filled traces only when there were at least 8 half hours of measured fluxes which passed all QA/QC criteria.

In order to measure soil CO$_2$ efflux (assumed to represent $R_e$) during winter, three opaque 10.1 cm diameter forced diffusion

chambers (eosFD, Eosense Inc., Dartmouth, NS, Canada) were installed ~50 m NW of the DL1 flux tower in a sandy, well-drained and exposed area with little soil organic matter. Measurements were available from August 18$^{th}$, 2018 to May 19$^{th}$, 2019. Implausible eosFD fluxes (21% of the time series) were excluded from analysis including those indicating uptake of CO$_2$ exceeding 0.5 $\mu$mol CO$_2$ m$^{-2}$ s$^{-1}$ and those associated with atmospheric or internal concentrations below 400 ppm which suggested problems with the analyzer's calibration, diffusion membrane or other factors related to gas transport through the frozen

soil-snow-atmosphere system. Daily total CO$_2$ emissions were calculated from the average of the remaining measurements.

CLASSIC runs were forced using 30-min meteorological observations at DL1. Measurements at DL1 included the four $R_n$ components downwelling and upwelling shortwave and longwave radiation (CNR1, Kipp & Zonen B.V., Delft, Netherlands), PA (PTB101B Barometer, Vaisala Oyj, Helsinki, Finland), $T_a$ and relative humidity (RH) at 1.5 m height (HMP-35C, Vaisala), $U$ and wind direction (propeller anemometer and wind vane at 1 m height; Wind Monitor, R.M. Young, Traverse City, MI,

USA), rainfall (tipping bucket rain gauge, TE525M, Texas Electronics, Dallas, TX, USA) and snow depth (sonic distance sensor, SR50-L, CSI). Solid precipitation was estimated from increases in snow depth over a 30 minute time step using a simple 1 mm water to 10 mm snow equivalent, when $T_a$ was below -2°C. Specific humidity was calculated as $q = \frac{e \cdot 0.622}{PA - e}$, where $e = \frac{RH}{100} \cdot e_s$ with the vapour pressure $e$ (Pa) and saturation vapour pressure $e_s$ (Pa). Measurements of $R_{SW}$, $R_{LW}$, PA, $T_a$, RH, $U$, and rainfall were gap-filled using duplicate sensors at nearby Daring Lake weather stations and flux towers located

within 2 km of the DL1 flux tower after adjusting for offsets. Any remaining gaps in PA, $T_a$, RH, and $U$ were filled using the nearest Environment Canada observations at Whatì (63°8.018'N, 117°14.684'W, 271.3 m asl) after regressing available observations with DL1 to adjust for differences in elevation and climate. Remaining gaps in $R_{SW}$ were filled with potential $R_{SW}$ calculated using DL1's latitude and longitude following Stull (1988) and gaps in $R_{LW}$ following Crawford and Duchon (1999) using gap-filled $T_a$ and an estimate of cloud cover based on the ratio of gap-filled observed $R_{SW}$ to potential $R_{SW}$. Precipitation

was obtained from its two components, rainfall and snow depth increments converted to snow water equivalent using a factor





of 10. Missing rainfall and snow depth increments were filled with Environment Canada Lupin station data (65°45.55'N, 111°15'W, 490.1 m asl) as these variables were not available from the Whatì station.

Additional weather observations at DL1 used in this study included up- and downwelling photosynthetically active radiation (PAR) (Quantum sensor, LI-190SA, LI-COR Inc.), soil temperatures (copper-constantan thermocouples) at 5, 25, and 60 cm
depths, volumetric soil water content (VWC) (water content reflectometer, CS615, CSI) at 7 and 20 cm depths (beginning in late August 2015 for the 20 cm depth) in a drier area representative of the majority of the tower footprint, and $G$ (soil heat flux plates at 7 cm depth, HFT3, CSI) adjusted to represent surface soil heat flux using the rate of change in energy stored in the layer of soil above the plates. PAR reflectivity was calculated as the ratio of upwelling and downwelling PAR.

Porosity, the liquid water content at wilting point and other hydraulic and thermal soil properties are determined in CLASSIC
using pedotransfer functions based upon the prescribed soil textures (Cosby et al., 1984; Clapp and Hornberger, 1978). For organic soils, peat type (fibric, hemic or sapric) dependent values are assigned to $\theta_p$ and wilting point following Letts et al. (2000). The modelled porosity and other soil properties did not necessarily correspond precisely to observed soil characteristics. For example, modelled $\theta_p$ for the surface organic layer and deeper mineral soil layers with 80% sand content was 0.93 and 0.39 m$^3$ m$^{-3}$, respectively, while observed $\theta_p$ for the top 10 cm and deeper soil layers in the field was 0.77 and 0.46 m$^3$ m$^{-3}$,
respectively. Although there may be differences in the absolute VWC (m$^3$ m$^{-3}$), the pore water held in the soil between field capacity and wilting point matric potentials are likely comparable. To facilitate this comparison, both observed and modelled VWC were scaled by their respective minimum and maximum values during 2004-2017 to produce a relative VWC value for each time step following

$$\text{VWC}(t) = (\theta(t) - \theta_{min})/(\theta_p - \theta_{min}). \tag{11}$$

**2.2.3 Bias-corrected reanalysis climate data**

A merged, reanalysis-based atmospheric forcing dataset (GSWP3-W5E5-ERA5) was bias corrected to the meteorological observations at DL1 and used to spin-up and drive CLASSIC for the historical simulation over 1901-2004, when site observations were not available. The 1901-1978 portion of the GSWP3-W5E5-ERA5 dataset was extracted from the Inter-Sectoral Impact Model Intercomparison Project 0.5° GSWP3-W5E5 atmospheric forcings (Kim, 2017; Lange, 2019, 2020a, b) and bilinearly
remapped to a 0.25° grid. The 1979-2018 period is based on the 0.25° ERA5 (ECMWF, 2019) time series that have been corrected so that long-term climatological means match those of the overlapping period of the GSWP3-W5E5 dataset.

Multivariate bias correction by N-dimensional probability function transform (MBCn) (Cannon, 2018) was used to adjust daily $R_{SW}$, $R_{LW}$, minimum and maximum $T_a$, precipitation rate, PA, $q$, and $U$ variables (1901-2018) from the GSWP3-W5E5-ERA5 data point nearest to DL1 to match the statistical characteristics — marginal distributions and multivariate dependence
structure — of the in situ observations. GSWP3-W5E5-ERA5 data at the DL1 grid cell were adjusted using the 2004-2018 observational period for calibration, with MBCn applied over 15-year sliding windows from 1901-2018. In each window, the central year is replaced, the window is slid one year, the forcings are bias adjusted using MBCn, etc. until the end of the GSWP3-W5E5-ERA5 dataset is reached. To ensure an unbiased seasonal cycle, adjustments were applied over 30-day sliding





intra-annual blocks of days. Outside of the 2004-2018 calibration period, changes in corrected quantiles were constrained to
match those in the GSWP5-W5E5-ERA5 dataset, i.e., the adjustments are trend-preserving (Cannon et al., 2015).

After bias adjustment, daily variables were temporally disaggregated to the required 30-min time step following the same
procedure as Melton and Arora (2016), where PA, $q$ and $U$ are linearly interpolated and $R_{LW}$ is uniformly distributed over the
day. Dependent on DL1's latitude and the day of the year (DOY), $R_{SW}$ and $T_a$ are diurnally distributed (adapted from Cesaraccio
et al., 2001). The daily amount of precipitation was used to determine the number of half-hours during which precipitation
occurred throughout the day (Arora, 1997) and was then randomly distributed over the wet half-hours (Melton and Arora,
2016).

### 2.2.4 Simulations

We performed three site-level simulations using different dominant PFT types (shrubs, grasses and trees) and one simulation
with the original $\beta$ formulation (Equation 4) with the shrub PFTs to illustrate the impacts of the model modifications on
simulated energy and C fluxes and to compare with observations made at DL1. The fractional coverage of the PFTs used in the
shrub, grass and tree simulations for the DL1 site are shown in Table 3. For the shrub simulation, the broadleaf evergreen and
broadleaf deciduous shrub and sedge cover reflect the vegetation observed at DL1. The grass simulation was set to $C_3$ grasses
with an equivalent total plant cover (60%). For the tree simulation, needleleaf evergreen trees and needleleaf deciduous trees
were chosen for the PFTs as the dominant tree species around treeline in the northern high-latitudes are black and white spruce
and larch (Sirois, 1992). For example, at DL1, which is only 70 km north of a diffuse treeline, sporadic clusters of stunted black
spruce are present at the base of sheltered south-facing slopes. Bare ground fractional coverage was 40% for all simulations,
although at DL1 most of the ground not covered by vascular plants is covered by mosses and lichen.

**Table 3.** Coverage of plant functional types (PFTs) used in the simulations.

| Simulation | PFT coverage | | | Bare ground |
|---|---|---|---|---|
| Shrubs (original and new $\beta$) | 30% Broadleaf Evergreen Shrubs | 12% Broadleaf Deciduous Cold Shrubs | 18% Sedges | 40% |
| Grass | | 60% C3 grass | | 40% |
| Trees | 30% Needleleaf Evergreen Trees | 12% Needleleaf Deciduous Trees | 18% C3 grass | 40% |

The top 10 cm soil layer was set as a fibric organic layer (see Letts et al., 2000) with the deeper layers set as mineral soil
consisting of 80% sand, 4.4 % clay and 3% organic matter (apart from the second layer, which was assigned 8% organic matter)
to best reflect average soil characteristics observed at DL1. The bias-corrected GSWP3-ERA5 dataset for 1901-1925 was used
repeatedly to drive the model until model C pools reached an equilibrium state, defined as a change in the annual C stocks of <
0.1%. The spin-up used the atmospheric $CO_2$ concentration from 1901 (Le Quéré et al., 2018). Starting from the equilibrium
model spinup state, a transient simulation was performed for the period 1901-2017 using time varying $CO_2$ concentrations, the
bias-corrected GSWP3-ERA5 meteorological forcing data for the years 1901-2003, and the meteorological observations from
DL1 for the years 2004-2017 as forcing data for CLASSIC.


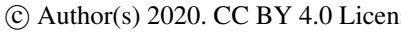


## 3 Results

### 3.1 Vegetation and soil carbon

The three model simulations using shrub, grass and tree PFTs produced vegetation with different characteristics (Table 4). Trees reached 20 m in height while shrubs and grass remained below 0.35 m. The simulated shrub height of 0.22 m was very similar to observations at DL1 (Table 4).


As expected, simulated stem and root biomass were much larger for trees than shrubs, especially stem biomass. Green leaf biomass was similar for shrubs and trees but smaller for grass. Compared to observations, simulated shrub leaf and stem biomass were 1.4 to 2 times too high, respectively, but were an appropriate order of magnitude. The simulated ratio of stem and leaf biomass was 1.4:1, while the observed ratio was nearly 1:1, although there was variability among the three sample plots (ratios varied from 0.7 to 1.15). LAI was overestimated by all three simulations, but was closest for the shrub simulation.


Each simulation produced detrital C pool (soil and litter C pools) estimates within the uncertainty bounds of the measured soil C at DL1, but again the shrub simulation was closest to the observed mean (Table 4).

**Table 4.** Vegetation and soil characteristics observed at the Daring Lake tundra (DL1) research site and modelled for this site using three simulations of CLASSIC with different plant functional types and the new ground evaporation efficiency parameterization. Observations of vegetation height, LAI and active layer depth at DL1 are described in the Methods section. Mean rooting depth was approximated from visual observations in the field. Biomass was assessed by harvesting all standing living vascular vegetation from three 0.25 m$^2$ plots, sorting by species and separating leaves and stems. Material was dried at 35°C to constant weight and converted to C assuming a 2:1 dry weight to C ratio. Soil C was assessed using loss on ignition and elemental C analysis of soil cores from 8 random soil pits.

| Characteristic | Model simulation | | | Observations |
|---|---|---|---|---|
| | Shrubs | Grass | Trees | |
| Max. vegetation height (m) | 0.22 | 0.35 | 20.73 | 0.18 ± 0.01 (SE) |
| Mean rooting depth (m) | 0.50 | 0.67 | 0.63 | ~0.40 |
| Max. LAI (m$^2$ m$^{-2}$) | 1.1 | 1.8 | 2.0 | 0.52 ± 0.05 (SE) |
| Max. green leaf biomass (g C m$^{-2}$) | 123 | 74 | 141 | 90 ± 7 (SE) |
| Max. stem biomass (g C m$^{-2}$) | 176 | 0 | 2199 | 85 ± 27 (SE) |
| Max. root biomass (g C m$^{-2}$) | 490 | 434 | 657 | - |
| Soil and litter C (kg C m$^{-2}$) | 17.3 | 21.7 | 15.3 | 18.5 ± 4.7 (SD) for 0-80 cm |
| Active layer depth (m) | 1.5 | 1.4 | 1.3 | 0.86 ± 0.03 (SE) |

### 3.2 Soil temperature and moisture

Simulated mean daily soil temperatures ($T_s$) of model layers 1 (0-10 cm depth), 3 (20-30 cm depth) and 6 (50-60 cm depth)


agreed well with field measurements between 2004 and 2017 with coefficients of determination ($R^2$) between 90 and 93% and





root-mean-square errors (RMSEs) between 2.2 and 2.5°C (Figures 1 and S3 and Table S3). However, in deeper soil layers, simulated $T_s$ was generally less seasonally variable than observations. As a result, simulated $T_s$ was slightly warmer in winter and slightly cooler in summer (Figure 1). This problem was exacerbated in the deeper layers with the original $\beta$ formulation (Equation 4; Figure 1). Differences in daily $T_s$ between the three simulations were small, but the shrub simulation showed

slightly higher $T_s$ year-round (Figure S3), especially for layers 3 and 6 and agreed slightly better with measurements during summer (June through August). For example, the RMSE for layer 3 (20-30 cm) $T_s$ was 1.6°C, 1.9°C and 2.1°C and for layer 6 (50-60 cm) was 1.5°C, 1.7°C and 1.9°C for the shrub, grass and tree simulations, respectively. RMSEs were larger in winter for all simulations with 2.7°C, 2.7°C and 2.6°C for layer 3 and 2.6°C, 2.6°C and 2.4°C for layer 6 for the shrub, grass and tree simulations, respectively. All three simulations overestimated active layer depth by 50-70% compared to the

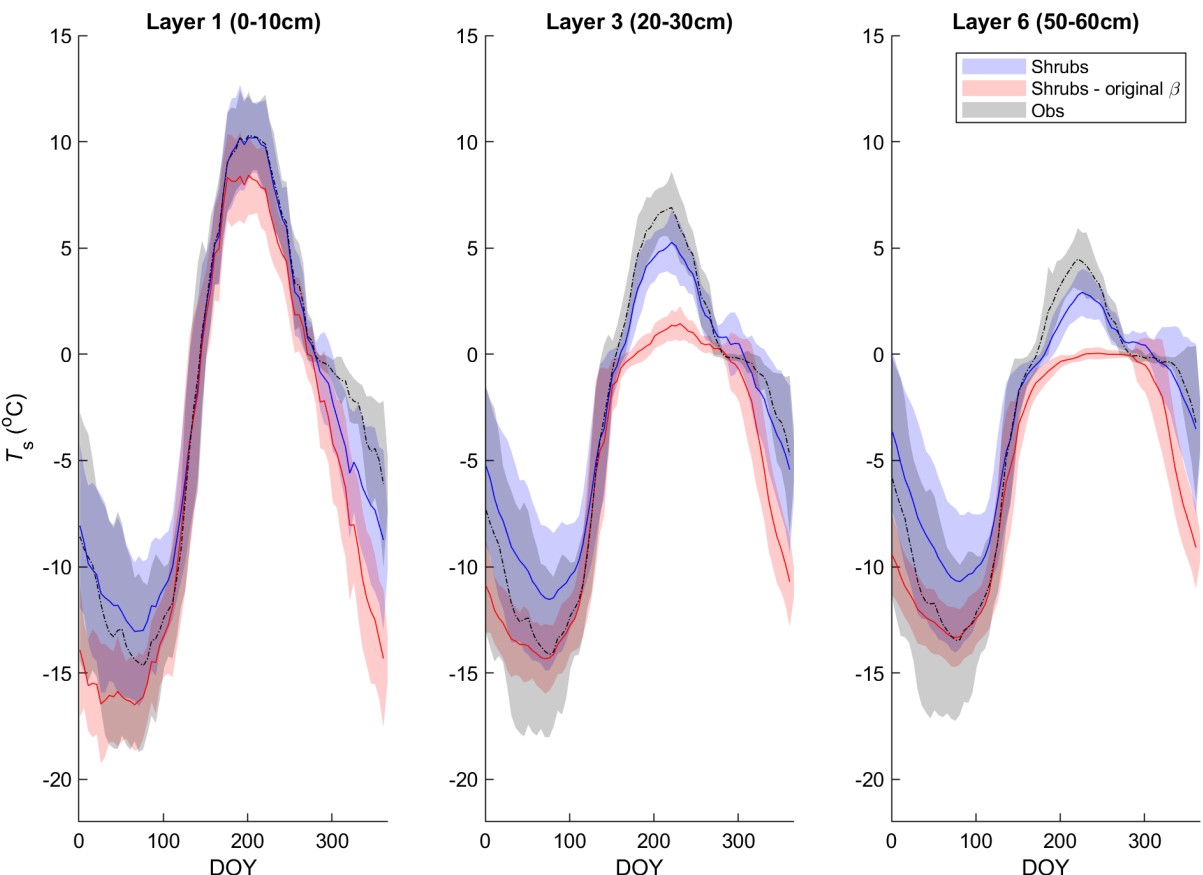

**Figure 1.** Mean 5-day average soil temperature for layer 1 (0-10 cm depth), 3 (20-30 cm depth) and 6 (50-60 cm depth) of the shrub simulations using the new (Merlin et al., 2011) and original $\beta$ (Lee and Pielke, 1992) formulation compared to measurements at 5, 25 and 60 cm depth, respectively, averaged over 2004-2017. Shaded areas show the standard deviation of the daily mean for 2004-2017.

average depth observed at DL1 (Table 4). Even though active layer depths vary spatially at DL1, they have not been observed





to exceed approximately 1.2 m. On average, simulated snow depth represented the observations well (Figure 2). However, the model tended to be snow-free earlier than observations by $3 \pm 4$ days (mean $\pm$ SD) with a range of 11 days earlier to 5 days later. Modelled mean daily relative VWC for the top soil layer (0-10 cm) averaged over 2004-2017 agreed reasonably well with

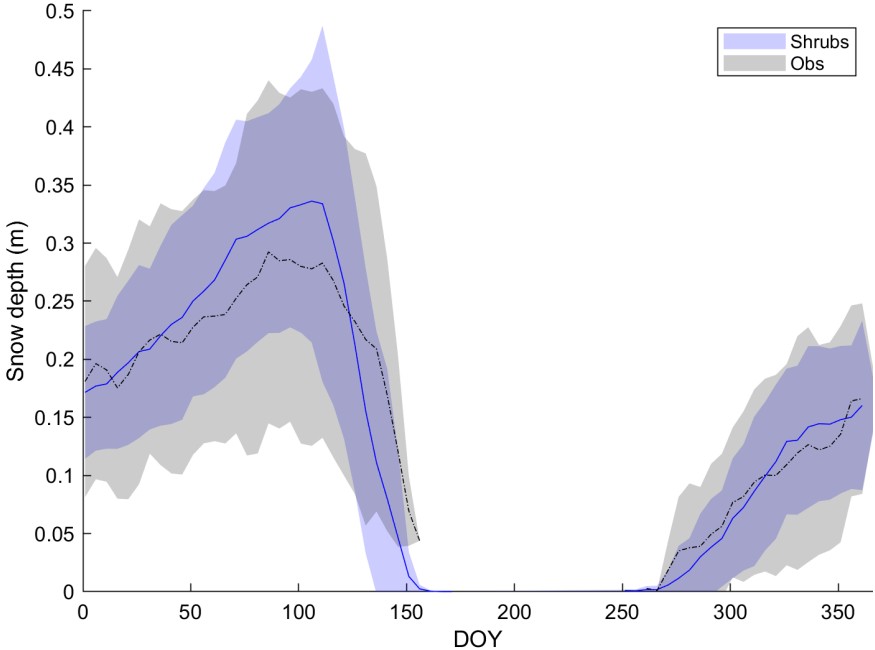

**Figure 2.** Mean observed and modelled 5-day average snow depth averaged over 2004-2017 for the shrub simulation. Shaded areas show the standard deviation of the daily mean for 2004-2017.

relative VWC measured at 7 cm depth (Figure 3a). Although interannual variability was high, VWC tended to be overestimated
by the model around snowmelt and underestimated later in the growing season (early August to mid-October) regardless of PFT simulation. At the end of the growing season, the shrub simulation represented VWC the best; modelled relative VWC was lower than observed by 0.09, 0.17 and 0.18 $m^3\ m^{-3}$ for the shrub, grass and tree PFT simulations, respectively. Simulated end of growing season relative VWC for the 10-20 cm layer was lower than observed by 0.11, 0.26 and 0.27 $m^3\ m^{-3}$ for the shrub, grass and tree PFT simulations, respectively. The original $\beta$ formulation, which was known to overestimate ground
evaporation (Sun and Verseghy, 2019), greatly underestimated relative VWC by on average 0.23 $m^3\ m^{-3}$ for the top soil layer and 0.46 $m^3\ m^{-3}$ for the 10-20 cm layer compared to observations throughout the growing season (Figure 3).

### 3.3 Turbulent energy fluxes (sensible and latent heat flux)

Differences in LE between the shrub, grass and tree PFT simulations were relatively small on average (Figure 4a). The adoption of the Merlin et al. (2011) $\beta$ formulation reduced overestimation of LE by approximately 30% during and just after snowmelt
(mid-May to mid-June) (Figure 4c). In summer, the new $\beta$ formulation greatly reduced variability in LE and ensured there

**Figure 3.** Mean 5-day average simulated relative volumetric water content (VWC, m$^3$ m$^{-3}$; see Equation 11) a) for the top model soil layer (0-10 cm depth) and observations at 7 cm depth and b) for the second layer (10-20 cm depth) and observations measured at 20 cm depth averaged over 2004-2017 for the shrub, grass and tree simulations. For the shrub simulation, results using the new (Merlin et al., 2011) and original $\beta$ (Lee and Pielke, 1992) formulation are shown. Measurements at the 20 cm depth were only available starting late August 2015, while measurements at the 7 cm depth began in June 2004. Shaded areas show the standard deviation for 2004-2017 for the model results and observations at 7 cm depth and for 2015-2017 for observations at 20 cm depth.

were no summer dates with unrealistically low LE (Figure 4c). However, all three PFT simulations with the new $\beta$ formulation still overestimated LE during and just after snowmelt and underestimated LE in summer (starting in late June). Average annual





LE was 514 MJ m$^{-2}$, 397 MJ m$^{-2}$, 430 MJ m$^{-2}$, and 434 MJ m$^{-2}$ (or 16.3 W m$^{-2}$ d$^{-1}$, 12.6 W m$^{-2}$ d$^{-1}$, 13.6 W m$^{-2}$ d$^{-1}$, and 13.8 W m$^{-2}$ d$^{-1}$ on average) for the observations, the shrub, grass and tree simulations, respectively. The observed annual value only

includes data from DOY 95-310 as observations were not available for the whole year, but model results suggest that LE during the missing time period likely contributed very little (<2%) to the annual total.

**Figure 4.** Mean 5-day average a) latent (LE) and b) sensible heat flux (*H*) over 2004-2017 for the shrub, grass and tree simulations using the new $\beta$ formulation (Merlin et al., 2011) and the original $\beta$ (Lee and Pielke, 1992) formulation (c) and d)) along with EC tower observations. Shaded areas show the standard deviation of the daily mean for 2004-2017.

ET as simulated by the shrub PFT with the new $\beta$ was dominated by ground evaporation (*E*) until mid-June, peaking shortly after snowmelt (Figure 5). Ground *E* was slightly higher for the grass simulation and slightly lower for the tree simulation in





spring compared to the shrub simulation (Figure 5). Transpiration ($T$) was an important contributor to ET from mid-June to
early September peaking in early August. Maximum $T$ was 40-45% greater for the grass and tree PFT simulations compared
to the shrub PFT simulation (Figure 5), resulting in slightly greater ET (also shown as greater LE in Figure 4a). For all three
simulations, $E$ of water intercepted by the canopy was a minor component of total ET.

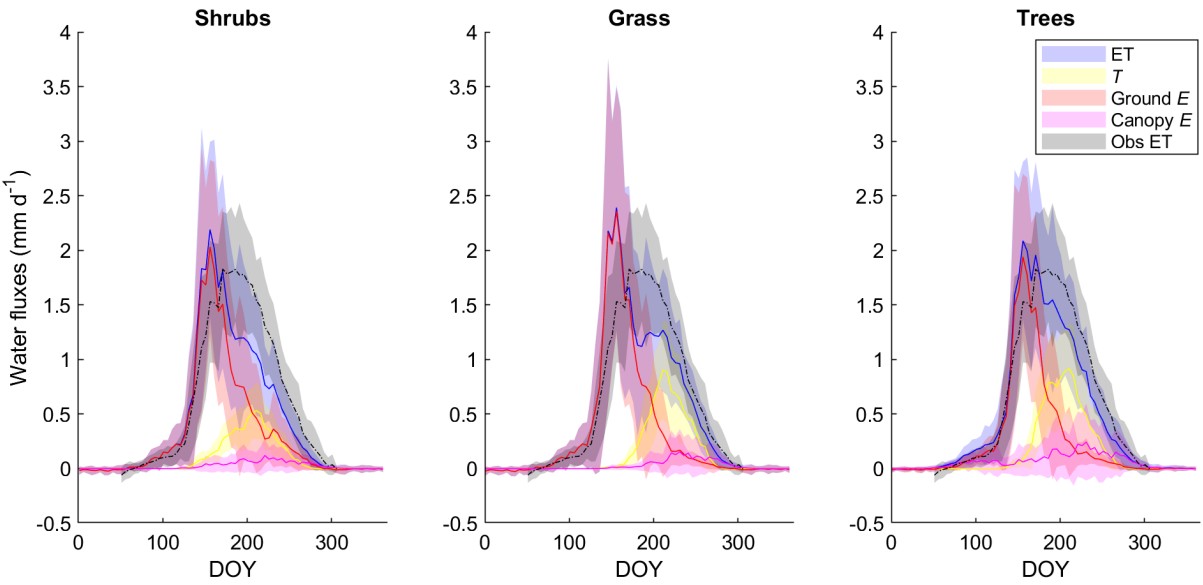

**Figure 5.** Mean 5-day average modelled evapotranspiration (ET) and its component fluxes (transpiration ($T$), ground and canopy evaporation
($E$)) along with observed ET averaged over 2004-2017 for the shrub, grass and tree PFT simulations using the new $\beta$ (Merlin et al., 2011)
formulation. Shaded areas show the standard deviation of the daily mean for 2004-2017.

For $H$, the shrub and grass simulations were similar, but the tree simulation greatly overestimated $H$ especially from mid-
April to the end of May (Figure 4b). The average annual total $H$ for the tree simulation (1046 MJ m$^{-2}$ or 33.2 W m$^{-2}$ d$^{-1}$) was
about 1.5 times as large as for the shrub (659 MJ m$^{-2}$ or 20.9 W m$^{-2}$ d$^{-1}$) and grass simulations (605 MJ m$^{-2}$ or 19.2 W m$^{-2}$ d$^{-1}$)
and more than 2.6 times the observed value (398 MJ m$^{-2}$ or 12.6 W m$^{-2}$ d$^{-1}$ for the period DOY 95-310). Only considering the
time period where measurements were available, average $H$ for the tree simulation was still 2.3 times the observed value. The
original $\beta$ formulation and shrub PFT simulation decreased $H$ throughout the year except in summer, so it had relatively little
impact on the annual $H$ (592 MJ m$^{-2}$ or 18.8 W m$^{-2}$ d$^{-1}$) compared with the new $\beta$ formulation (659 MJ m$^{-2}$ or 20.9 W m$^{-2}$ d$^{-1}$).
These large differences in spring $H$ among PFT simulations could be linked to differences in simulated $R_n$ (Figure S6) and
albedo (not shown). Albedo and $R_n$ were similar for the shrub and grass simulations with average albedo values of 0.91 and
0.92, respectively, during late winter, which was only slightly lower than the observed value of 0.97. The tree simulation,
however, had a much larger $R_n$ and lower albedo (0.55) than the shrub and grass simulations during winter, as the tall trees
cannot be buried by snow.



## 3.4 Net ecosystem CO$_2$ exchange and its component fluxes


The shrub PFT simulation with the new $\beta$ formulation best represented observed NEP and its component fluxes (Figure 6 and Table 5). The new $\beta$ formulation raised modelled VWC, reduced water stress and supported GPP rates that closely resembled GPP derived from field observations at DL1. In contrast, the original $\beta$ formulation resulted in GPP values that were only 26% of observed values (Table 5 and Figure S4). NEP simulated with the shrub PFT and new $\beta$ formulation nevertheless was

overestimated in spring as GPP started approximately 9 days earlier than observed resulting in total growing season uptake that was 43 g C m$^2$ larger than observations (Table 5). In contrast, simulated total growing season $R_e$ agreed well with observations (Table 5) although it was slightly higher than observed in spring and lower in summer (Figure 6).

NEP simulated with the grass PFT lagged observations in the spring, peaking on average 49 days after the observations (Figure 6a). Summer maximum GPP for the grass simulation was higher and, on average, reached a daily maximum 9 days

later than observations. Simulated grass PFT GPP continued late into the fall, when observed GPP had declined to near zero. Total $R_e$ was similar (Table 5) to $R_e$ simulated with the shrub PFT, but again seasonal trends were offset by on average 10 days.

For the tree PFT simulation, simulated NEP and GPP were reasonably close to the measured values in the spring, although the start of NEP uptake was still about 10 days early (Figure 6). Uptake was overestimated in mid-summer and into the fall compared to measurements, resulting in the largest growing season NEP (Table 5). Both chamber and EC measurements show

**Table 5.** Mean ± SD annual and growing season (GS; May 1 - September 30) net ecosystem productivity (NEP), gross primary productivity (GPP) and ecosystem respiration ($R_e$) averaged over 2004-2017 for the shrub simulations using the new (Merlin et al., 2011) and original $\beta$ (Lee and Pielke, 1992) formulation, for the grass and tree simulations, and for observations. Observations were only available during the growing season. Standard deviations (SD) for the observed and simulated fluxes are calculated by error propagation of the SD of daily values.

| Simulation | NEP [g C m$^{-2}$] | | GPP [g C m$^{-2}$] | | $R_e$ [g C m$^{-2}$] | |
| --- | --- | --- | --- | --- | --- | --- |
| | Annual | GS | Annual | GS | Annual | GS |
| Observations | - | 12 ± 5 | - | 214 ± 7 | - | 202 ± 5 |
| Shrubs | -18 ± 4 | 55 ± 4 | 276 ± 6 | 273 ± 6 | 294 ± 3 | 218 ± 3 |
| Grass | -15 ± 6 | 61 ± 6 | 279 ± 9 | 271 ± 8 | 295 ± 4 | 210 ± 3 |
| Trees | -25 ± 5 | 82 ± 5 | 374 ± 8 | 365 ± 8 | 399 ± 4 | 283 ± 4 |
| Shrubs - original $\beta$ | -7 ± 2 | 9 ± 2 | 57 ± 2 | 56 ± 2 | 64 ± 1 | 47 ± 1 |

a more rapid decrease in CO$_2$ emissions throughout September than shrub PFT simulations, as soil and air $T$ dropped to or below 0°C (Figure 6). During the winter, all three simulations had lower NEP and higher $R_e$ than observed fluxes from the chambers and EC system (Figure 6). However, the observations have some significant uncertainties as fluxes were measured over one winter (fall 2018 - spring 2019) only, which was outside the simulation period of 2004-2017. In addition, forced diffusion chambers have a much smaller footprint (41 cm$^2$) with less diverse ground cover than the EC footprint of 1 ha or

more.



**Figure 6.** Mean 5-day average a) net ecosystem productivity (NEP), b) gross primary productivity (GPP), c) ecosystem respiration ($R_e$) for the shrub, grass and tree simulations alongside EC tower based observations. d) Simulated $R_e$ broken down into its component fluxes, autotrophic ($R_a$) and heterotrophic respiration ($R_h$), for the shrub simulation averaged over 2004-2017. Observed NEP (a) and $R_e$ (c and d) include both EC measurements during the growing season averaged over 2004-2017 as well as chamber measurements made between August 2018 and May 2019. Shaded areas show the standard deviation of the daily mean for 2004-2017.

During summer, the shrub simulation's $R_a$ and $R_h$ were roughly half of $R_e$ (Figure 6d). Fall and winter $CO_2$ emissions were primarily through $R_h$ although $R_a$ remained above zero. Otherwise, $R_a$ closely followed GPP trends. Similar patterns with slightly larger values were observed for the tree simulation while the grass simulation's $R_a$ was near zero for the winter months (Figure S7).



All three model simulations suggest that DL1 was a net source of $CO_2$ over the 2004-2017 period (Figure 7 and Table 5). Winter and shoulder season $CO_2$ emissions from October to April exceeded May through September $CO_2$ uptake by 26-32% on average for the three simulations. The magnitude of winter $CO_2$ loss differed among simulations with a net loss of 211 g C m$^{-2}$ for grass, 254 g C m$^{-2}$ for shrubs and 344 g C m$^{-2}$ for trees over the 14 year period, which was equal to an average annual net $CO_2$ emission of 15-25 g C m$^{-2}$ yr$^{-1}$ (Table 5). With the caveats discussed above regarding combining chamber and EC data streams, annual net $CO_2$ loss observed at DL1 was 17 g C m$^{-2}$ yr$^{-1}$ (7 $\pm$ 5 g C m$^{-2}$ for DOY 77-309 and 10 g C m$^{-2}$ for DOY 1-76 and 310-365).

    The simulations of NEP differed in response to interannual variability in meteorological forcing (Table S2). On average, the 2010-2017 growing season was 2.1°C warmer and over two times wetter than 2004-2009, respectively (t-test p < 0.05) (Table S2). NEP simulated using the grass PFT was more sensitive to these different weather conditions and thus more variable than the other two simulations over the study period including a few years (2006, 2011, 2012, 2013, 2016 and 2017) where DL1 was simulated to be a net $CO_2$ sink. Annual net $CO_2$ uptake was also simulated for a few years (2011, 2012, 2013 and 2017) using the tree PFTs, while the shrub simulation only showed annual net $CO_2$ uptake in 2012.

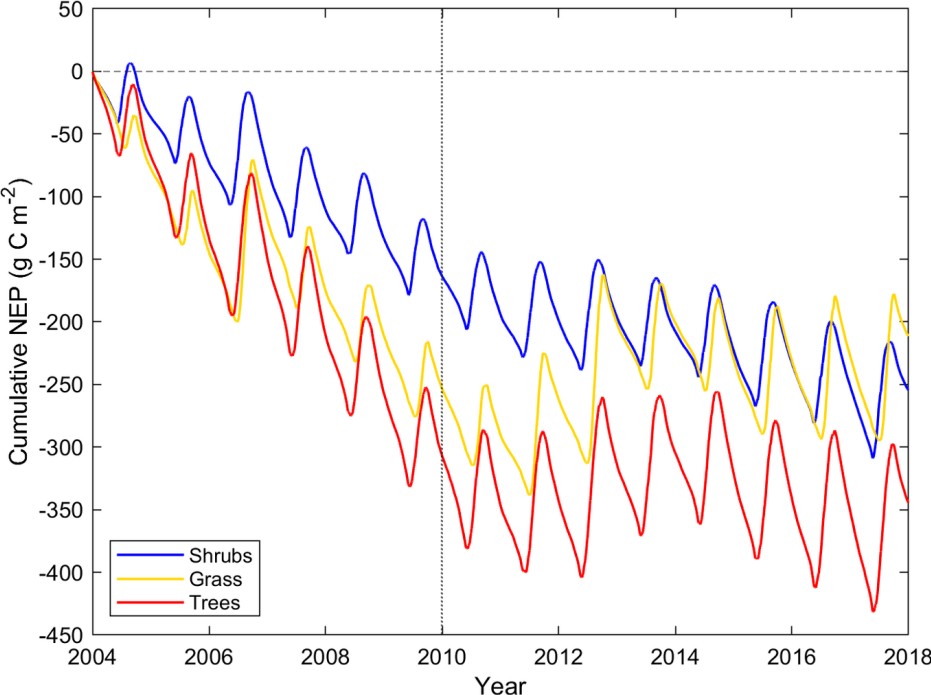

**Figure 7.** Cumulative daily modelled net ecosystem productivity (NEP) for 2004-2017 for the shrub, grass and tree simulations. Negative values indicate the land surface was releasing $CO_2$ into the atmosphere. The vertical dotted line indicates the year marking a shift in weather with colder and drier weather before 2010 and warmer and wetter weather thereafter.



## 4   Discussion

Although we focus on high-latitude shrubs, shrubs are an important growth form in multiple regions and biomes, cover about
40% of the land surface including polar and alpine tundra, arid regions, and wetlands and are often dominant within forest
understories (Götmark et al., 2016). Shrubs, as a growth form, have a number of advantages compared to small trees. For
example, in disturbed and low-productivity areas, shrubs have higher growth rates and having multiple, short or bendable
stems, are more resilient to storm damage, weight of snow loading, and can more readily recover from stem breakage and thus
have higher survival rates under extreme conditions (Götmark et al., 2016).

This study highlights improved simulations of surface-atmosphere interactions at a low Arctic upland tundra site in Canada
with the introduction of shrub and sedge PFTs and an improved parameterization of ground evaporation within CLASSIC. We
compare these results to other field and model studies that highlight key tundra ecosystem processes and the impacts of shrubs
on evaporation, soil thermal regimes and C cycling and storage.

### 4.1   Tundra-atmosphere water vapour exchange

This study addressed a high soil evaporation bias that has also been observed in other models' simulations of grasslands, wet-
lands and forests, but especially in regions with sparse vegetation (Sun and Verseghy, 2019; Decker et al., 2017; Kauwe et al.,
2017; Mu et al., 2020). Merlin et al. (2011)'s empirical formulation of the soil evaporation efficiency reduces simulated high
rates of ground evaporation, particularly in spring, which avoided overdrying the 0-10 cm soil layer and greatly underesti-
mating summer soil temperature, LE and GPP, particularly in years with less summer rainfall. Although there were no field
measurements to distinguish the soil moisture in the top few cm from the bottom few cm of this soil layer at DL1, it is clear
from the high VWC at 20 cm depth that soils below the surface remain moist throughout the summer. As DL1 is located on a
shallow slope below an esker, just west of several water tracks (channels of high moisture content in permafrost dominated soils
through which water is routed downslope; see Curasi et al. (2016) for a detailed description of these tundra features), which
lead to a sedge wetland, it is expected that the DL1 site receives and sheds water through lateral flows, which CLASSIC does
not simulate. Accordingly, Grant et al. (2015) found that lateral surface and subsurface flows were needed to model seasonal
soil moisture variations at DL1 using the ecosystem model *ecosys*.

Even with the new $\beta$ formulation, ground $E$ at the time of snowmelt remained overestimated, likely due to a lack of infiltration
into the porous surface soil. CLASSIC simulated ponded water during and up to ~12 days after snowmelt resulting in saturated
near-surface soil layers. In reality, water infiltrates into the soil faster with little ponding observed, as confirmed by repeat
photography in the field. The ponding that did occur was generally within microtopographic depressions and for a shorter time
period of about a week or less following snowmelt.

Merlin et al. (2011)'s empirical formulation of the soil evaporation efficiency can be adapted for different thicknesses of the
top layer by using different soil layer thickness- and soil texture-dependent values of the exponent in Equation 5. Using Merlin
et al. (2011)'s formulation, CLASSIC's 10 cm thick top layer, however, was not able to represent a thin, dry surface layer
reducing ground $E$. Attempts to reduce the top soil layer to 5 cm thickness or less created instabilities in the model. Excessive





ground $E$ was also observed in the Community Land Model (CLM), especially for sparse canopies or bare soil areas (Swenson and Lawrence, 2014). In order to address this issue, Swenson and Lawrence (2014) implemented a new soil resistance parameterization in the CLM version 4.5 that includes a dry surface layer whose thickness is determined from the moisture in the top soil layer and where evaporation is determined by water vapour diffusion through this layer. Using this parameterization

in CLM resulted in less bias in ET, as soil $E$ decreased due to higher resistances, even when soils were moist below the surface (Swenson and Lawrence, 2014). Another approach that could be employed in the future to prevent excessive ground $E$ in CLASSIC is a representation of a litter layer increasing resistance to water vapour and heat exchange at the soil surface (e.g., Mu et al., 2020; Decker et al., 2017).

At DL1, the true proportion of total growing season ET partitioned to $T$ is not known. However, a mini-lysimeter study was

carried out during the last week of July 2019 to quantify ET partitioning. Four pairs of 25 cm diameter and ~20 cm deep cores with vascular vegetation intact and vascular vegetation clipped at the ground surface were installed in pots set into the ground and reweighed. On average, $T$ was $54 \pm 29\%$ (SD) and $43 \pm 25\%$ of ET after one and two days, respectively. Given the high variability among lysimeters, there was no evidence that $T$ was significantly different than 50% of total ET (t-test p=0.71 and 0.46, respectively). CLASSIC also simulated ET to be $50 \pm 22\%$ $T$ for the last week of July 2004-2017 using the shrub PFT

simulation with the new $\beta$ formulation. The contribution of $T$ to total ET for this period increased with grass ($64 \pm 22\%$) and tree ($69 \pm 22\%$) PFT simulations. Intercomparisons of ESMs highlight that spatial variations in $T$:ET are largely driven by LAI, but many of these models underestimate the globally averaged ratio of $T$ to ET compared to observations (Lian et al., 2018; Chang et al., 2018). Lian et al. (2018) suggested that model deficiencies in canopy light use, interception losses and root water uptake processes contributed to the underestimation of $T$. Chang et al. (2018) found that inclusion of lateral flow and an

improved representation of water vapour diffusion within the soil reduced $E$ and increased $T$:ET, in agreement with Swenson and Lawrence (2014)'s findings.

Because $R_n$ was well represented but LE was underestimated in summer, the remaining energy had to be partitioned to other energy sinks. For the shrub PFT simulation with the new $\beta$ formulation, there was an overestimate of thaw depth and growing season $H$. Nevertheless, soil temperatures in this permafrost affected soil were reasonably well represented despite some

dampened response at depth resulting in soils that are slightly warmer than observed in winter and colder in summer. Melton et al. (2019) showed that using a large number of soil layers (20) and greater depth of ground layers (61.4 m) than the 3 soil layers in previous model versions improved the simulation of circumpolar ground temperatures and active layer depths in permafrost regions.

Due to trees remaining unburied by snow through the winter, the tree simulation did not represent $R_n$ or $H$ well during

late winter. This shows the importance of accurately representing burial of vegetation by snow and/or representing vegetation dynamics in the model such as shrub expansion and northward migration of the tree line in order to predict current and future energy feedbacks to the climate system (e.g., Chapin et al., 2005). Recent studies have also noted the potential contribution of atmospheric moisture (through increased ET) feedbacks on regional warming of circumpolar regions through shifts in vegetation (Pearson et al., 2013; Bonfils et al., 2012; Lawrence and Swenson, 2011). Total annual $T$ and ET over 2004-2017

was greater in our tree PFT simulation ($56 \pm 3$ mm ($\pm$ SD) and $162 \pm 7$ mm, respectively) over shrubs ($34 \pm 2$ mm and





± 8 mm, respectively). However, shrub $T$ and ET was not greater than grass PFT $T$ and ET (47 ± 3 mm and 164 ± 9 mm, respectively). Lafleur and Humphreys (2018) found there was little difference in growing season ET between DL1 and two neighbouring tundra sites with greater cover and height of shrubs. ET may not have differed among these and other tundra sites (McFadden et al., 1998, 2003), potentially due to compensation between ground $E$ and $T$. In this study, ground $E$ decreased

slightly in the tree PFT simulation compared to the shrub PFT simulation, as $T$ increased. However, further improvements to the model representation of the processes governing ET and additional evaluation with field data will improve our ability to characterize and quantify this potential climate feedback.

### 4.2   Shrub tundra CO$_2$ fluxes

Terrestrial ecosystem model estimates of annual CO$_2$ fluxes are especially useful in regions where year-round measurements

are rare and difficult to obtain as they present a means to quantify the annual C budget of a region. Natali et al. (2019)'s recent compilation of CO$_2$ fluxes from over 100 high-latitude sites from the Arctic and boreal northern permafrost region during the winter season (October-April) confirmed that tundra ecosystems emit substantial amounts of CO$_2$ through the winter months. Of the three methods, the rate of CO$_2$ emissions throughout the winter and shoulder seasons at DL1 was least with the chamber method and greatest in the model simulations. As noted earlier, differing scales of observation from < 1 to 100s of square

meters for chambers and EC techniques, among other methodological differences, present issues when trying to compare results between the two methods and creating a continuous record of observations, particularly in ecosystems where fluxes are small and susceptible to methodological bias (Campioli et al., 2016). When compared for overlapping time periods (September 2018, not shown), chamber measurements of CO$_2$ fluxes were lower than the EC-derived $R_e$. Some of this difference may be due to the method used to partition NEP into $R_e$ and GPP. Later in the season, as GPP declined to near zero with cold and

snow-covered conditions and EC NEP was equal to -$R_e$, chamber $R_e$ remained less than EC $R_e$. This difference was likely due to limited vegetation growth and survival within the chambers which remained in place and closed through the summer and winter. The three chambers were also located in well-drained sandy soils with minimal surface organic matter and thus did not represent the full tower footprint heterogeneity. For example, Campeau et al. (2014) reported a range in 0-50 cm soil organic C between 10 to 29 kg C m$^{-2}$ for the different vegetation communities characteristic of the DL1 flux tower footprint. Assuming

$R_e$ is influenced strongly by vegetation productivity, soil substrate and soil temperature, among other drivers (Virkkala et al. 2018), it is reasonable to expect spatial variation in $R_e$.

CLASSIC NEP and $R_e$ agreed well with EC observations through the fall and early winter, which was expected given that soil temperatures and detrital C stocks (soil and litter C) were well represented by the model. However, simulated $R_e$ was on average 0.25 g C m$^{-2}$ day$^{-1}$ larger than EC-estimated $R_e$ during late winter and early spring, which was likely a result of warmer

modelled winter soil temperatures. This overestimation of winter $R_e$ and its component fluxes, $R_a$ and $R_h$ was higher for the tree than the shrub and grass PFT simulations. Both $R_a$ and $R_h$ contributed through the winter as a result of maintenance respiration of greater stem and root biomass and higher base soil respiration rates despite slightly cooler soil temperatures (Figure S3). In tundra environments, vegetation traps drifting snow such that areas with taller and more abundant vegetation tend to have deeper snow and warmer winter soils (Sturm et al. 2005). CLASSIC does not include snow redistribution processes, which



may have limited the differences in simulated winter soil temperatures among PFTs and would limit differences in related C
cycle processes.

The ratio of winter $CO_2$ emissions to growing season uptake was tightly constrained between 1.26-1.32 for our simulations
at DL1, which was lower than Natali et al. (2019)'s estimate of ~1.61 for the northern Arctic and boreal permafrost region
(growing season was defined here as May-September in accordance with Natali et al., 2019). This may be due to regional
differences, overestimation of growing season uptake in our simulations as well as uncertainties in the growing season uptake
estimates from the process-based models they used. Our simulated winter emissions of 74-107 g C m$^{-2}$ y$^{-1}$ at DL1, however,
were within the range reported by Natali et al. (2019).

During and just after snowmelt, GPP was overestimated by CLASSIC for the shrub simulation and, to a lesser extent, the
tree simulation. Commane et al. (2017) noted that this is a common problem in Earth System Models where most CMIP5
models simulated net $CO_2$ uptake earlier in spring than observed, contributing to an overestimation of annual net $CO_2$ uptake.
In CLASSIC, this overestimation was likely due to a combination of simulated snowmelt occurring too early in the model,
and shrub productivity increasing too quickly after snowmelt completed. One possible reason for GPP overestimation in spring
is that CLASSIC does not account for effects of pigments such as anthocyanin, which are produced by evergreen shrubs
during fall and spring and peak shortly after snowmelt. Anthocyanin pigments cause reddening of the leaves and prevent
photodamage to photosynthetic tissues when radiation levels are high but $T_a$ and $T_s$ can be low (Oberbauer and Starr, 2002;
Wyka and Oleksyn, 2014). Pigment concentrations have been measured to be elevated in shrubs in higher light environments
(e.g., little snow cover, upward facing leaves oriented towards the sky or younger leaves with higher nitrogen content), resulting
in lower photosynthetic capacity than in greener leaves (Oberbauer and Starr, 2002). CLASSIC does not simulate leaf reddening
of evergreen shrubs in the spring, which could contribute to a high GPP bias in the model. Overestimation of GPP in the
spring could also be related to the representation of phenology such as timing of leaf onset, especially for deciduous shrubs.
The inclusion of a chilling requirement, in addition to growing degree days, has been shown to improve the timing of the
initiation of C uptake, and thus improve $CO_2$ flux simulations in other models, e.g. for Northern Alaskan deciduous vegetation
especially during warm springs (Jeong et al., 2012; Shi et al., 2020). The limited availability of budburst measurements in
tundra ecosystems, and significant changes expected due to climate warming, make improving modelled tundra phenology
challenging (Diepstraten et al., 2018). Finally, the addition of the nitrogen cycle in CLASSIC may help reduce a high GPP
bias as Low Arctic tundra plant growth is typically limited by nitrogen and phosphorus (e.g.,  Wieder et al., 2015; Wild et al.,
2018).

Regardless of PFT simulation, CLASSIC simulated DL1 as a net source of $CO_2$ over the 14 year period. Although there was
year-to-year variability in the magnitude of loss including some years with net $CO_2$ uptake for the three simulations, average
annual net $CO_2$ losses were not too different at 15 (grass PFT), 18 (shrub PFT) and 25 g C m$^{-2}$ yr$^{-1}$ (tree PFT). These net $CO_2$
loss estimates were similar to the combined EC and chamber estimate of 17 g C m$^{-2}$ yr$^{-1}$ as the overestimation of growing
season net $CO_2$ uptake in the model was compensated by an overestimation of winter $CO_2$ loss. Grant et al. (2011) using the
*ecosys* model, which included downslope lateral water flow (Grant et al., 2015), found DL1 to be an annual C sink using *ecosys*
with net uptake of 17-45 g C m$^{-2}$ yr$^{-1}$ from 2004-2007, as modelled $CO_2$ losses between September 1 and May 14 were lower



(24-31 g C m$^{-2}$) than net CO$_2$ uptake of 41-76 g C m$^{-2}$ from May 15-August 31. However, in agreement with our CLASSIC results, other permafrost-affected Arctic tundra ecosystems have also been observed and modelled as net sources of CO$_2$. For example, DL1's dwarf shrub tundra had similar net CO$_2$ losses as a heath tundra ecosystem in Alaska which lost on average 20 g C m$^{-2}$ y$^{-1}$ or 158 $\pm$ 53 g C m$^{-2}$ over an 8 year period where soils consistently warmed (EC observations; Euskirchen et al., 2017). In that same region, wet sedge tundra was observed to lose 668 $\pm$ 83 g C m$^{-2}$ (or 84 g C m$^{-2}$ y$^{-1}$) over 8 years (EC

observations; Euskirchen et al., 2017). Further south in Alaska, thaw depth was observed to deepen over six years in moist acidic tundra, which lost a similar amount of CO$_2$ on average, 87 $\pm$ 17 g C m$^{-2}$ (EC observations; Celis et al., 2017) while in contrast, the sedge dominated Zackenberg fen of Northeast Greenland was reported to be a net CO$_2$ sink of 50 g C mm$^{-2}$ y$^{-1}$ over a 10 year period although one year had an annual loss of 21 g C m$^{-2}$ yr$^{-1}$ (EC observations combined with process model; López-Blanco et al., 2020).

In our study, the grass PFT simulation was more sensitive to environmental conditions, primarily through variations in growing season net CO$_2$ uptake (Figure 7). In Euskirchen et al. (2017)'s study noted above, the wet sedge tundra's annual net CO$_2$ exchange was also more variable than tussock and heath tundra, but in that study, there was both less uptake and more loss during the growing and winter seasons, respectively, as soils warmed over time. Simulated shrub PFT NEP was less variable than observations at DL1 suggest. In a previous study at DL1, considerable variability in NEP from mid-May to the

end of August strongly correlated to summertime ecosystem-level photosynthetic capacity, which could reflect differences in the amount and/or productivity of photosynthetic tissues through variations in nutrient availability, overwinter tissue damage, etc. (Humphreys and Lafleur, 2011).

## 5   Conclusions

CLASSIC's newly implemented shrub and sedge PFTs improved representation of soil temperatures, soil moisture, CO$_2$ and

energy fluxes for a dwarf shrub tundra ecosystem, which was modelled to be a net CO$_2$ source over the 14-year study period. However, the timing of the onset of net CO$_2$ uptake in the spring was too early and contributed to the overestimation of growing season GPP and NEP. This issue may be resolved by incorporating leaf pigment dynamics and testing CLASSIC's new nitrogen cycle module (Asaadi and Arora, 2020) at the site. Another remaining issue is the overestimation of ET, particularly during the snowmelt period. The modified evaporation efficiency parameterization in CLASSIC was critical to represent observed soil

temperatures and reduce soil drying that otherwise resulted in water stress limitations later in the growing season. However, ground evaporation and total ET remained high, which might be resolved by implementation of a dry surface soil or litter layer in the future.

    Simulations of energy and CO$_2$ fluxes using shrub, grass and tree PFTs demonstrated the importance of representing tundra ecosystems with the appropriate PFTs in Earth System Models. The results of this study highlighted several important

processes influenced by PFTs. For example, burial of vegetation by snow had a substantial impact on early spring radiative and turbulent energy fluxes, PFT phenology and growth strategies influenced the timing and magnitude of net CO$_2$ uptake rates, and differences in C stocks along with surface energy exchange influenced the magnitude of winter CO$_2$ emission rates.



*Code and data availability.* The CLASSIC code v1.0.1 including shrub and sedge plant functional types is archived on Zenodo (Meyer et al., 2020a) and the eddy covariance and meteorological measurements made at the Daring Lake dwarf-shrub tundra site (DL1) between 2004

and 2017, which were used to drive and validate CLASSIC are available on Zenodo as well (Meyer et al., 2020b).

*Author contributions.* G.M. tested and improved the parameterization of the new CLASSIC PFTs, performed the model simulations, and wrote the manuscript. E.R.H and J.R.M. planned this study, helped set up the modelling framework and initialization files, and contributed to the analysis and manuscript. E.R.H. and P.M.L. made the measurements at DL1 and processed the observation data. A.J.C. bias corrected the GSWP3-W5E5-ERA5 reanalysis data for the DL1 site. All authors contributed to the final manuscript.

*Competing interests.* There is no conflict of interest.

*Acknowledgements.* This research was funded through the Natural Sciences and Engineering Research Council of Canada and Polar Knowledge Canada. We thank Steve Matthews, Karin Clark and the Daring Lake Terrestrial Ecosystem Research Station staff for logistical support, Michael Treberg for field and technical assistance and graduate and undergraduate students for field assistance.



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
