# Peer review of "Simulating shrubs and their energy and carbon dioxide fluxes in Canada's Low Arctic with the Canadian Land Surface Scheme Including biogeochemical Cycles (CLASSIC)"

_Biogeosciences, 2020_

## Referee Comment (RC1) · Anonymous Referee #1 · 15 Feb 2021

General comments:

The manuscript describes modifications to the Canadian Land Surface Scheme Including biogeochemical Cycles (CLASSIC) model, and it evaluates simulations of surface-atmosphere carbon dioxide ($CO2$), water, and energy exchange against observations from eddy covariance and carbon flux chambers for a dwarf-shrub tundra site in Canada's Southern Arctic ecozone. The study added new shrub plant functional types (PFTs) to the physical and biogeochemical sub-modules of the CLASSIC model. Additionally, the CLASSIC model was modified to correct for a high ground evaporation

bias which affects simulated energy fluxes and water-stress impacts on plant productivity. Simulations of the CLASSIC model were run to estimate annual and seasonal $CO_2$ fluxes, and model estimates were compared for three parameterizations (shrub, tree, and grass PFTs) and evaluated against observations. The ground evaporation bias correction, implemented as a new formulation of the coefficient B (used in the calculation of specific humidity at the surface), improved agreement between simulated and observed values for several variables (e.g., soil moisture in the upper soil layers, soil temperature in surface and deeper soil layers, spring and summer latent heat flux (LE), and $CO_2$ fluxes). When simulations with the new shrub PFTs were compared against simulations with grass or tree PFTs, the shrub simulations had the best agreement with observations for variables such as leaf area index (LAI), soil and litter carbon pools, vegetation height, mean rooting depth, summer soil temperature, and late growing season soil moisture. Notably, the shrub PFT improved simulation of vegetation burial by snow, which impacts net energy fluxes, phenology, and the timing and magnitude of net $CO_2$ fluxes. Together, the evaporation bias correction and new shrub PFTs led to improved agreement with observations considering the magnitudes of both net and component $CO_2$ fluxes. Results demonstrated the importance of incorporating appropriate PFTs in process-based models, particularly in terms of capturing the physical and biogeochemical impacts of shrubs in tundra ecosystems.

The manuscript is well written and well organized, and the lack of tundra-specific PFTs in the CLASSIC model is clearly framed as a problem to be addressed. The presented figures and tables provide clear evidence that the modifications to the CLASSIC model (both the addition of shrub PFTs and the correction of the ground evaporation bias) led to improved estimates of carbon and energy fluxes at the dwarf-shrub tundra site. The manuscript also acknowledged continuing modeling challenges, including overestimated net carbon uptake in the early growing season, overestimated winter $CO_2$ loss, and overestimated evapotranspiration during snowmelt (even with the new B formulation), and it provides suggestions for future model modifications that can address these challenges. The authors might want to consider elaborating further on any significant

trends present in the observed or modeled data over the 2004-2017 study period. For example, it would be helpful to report whether there were significant positive trends in growing season vegetation productivity, potentially indicating increases in shrub growth over the simulation period. Additionally, were temporal trends present in annual respiration, soil temperature, active layer depth, growing season length, or other variables? If significant trends are present, do they provide further information about physical or biogeochemical processes occurring at the site over the study period? Overall, the manuscript presents important modifications to the CLASSIC model that help to improve understanding of the drivers of $CO_2$ flux dynamics in shrub-dominated tundra ecosystems.

Specific comments:

Line 42: It might be worthwhile to mention that some emissions observed during the winter could be driven by the diffusion of stored $CO_2$ that was produced during the non-winter period (although the magnitude of the contribution is not known, e.g., Natali et al. 2019).

Lines 101-103: It sounds as if respiration processes Ra and Rh are determined as a function of C pools within the biogeochemical sub-module (i.e., daily timestep). Are daily respiration rates also determined as functions of soil temperature or other inputs? (E.g., Lines 514-515). What determines the base soil respiration rate (e.g., Line 517)? Is Ra estimated as a function of daily carbon assimilation?

Do the model parameters involved with estimating respiration vary by PFT and/or soil layer? It would also be helpful to briefly describe how changes to soil carbon pools are calculated in the different soil layers.

Line 111: Perhaps mention why these particular PFTs were selected to be added to the biogeochemistry sub-module? I.e., cold broadleaf deciduous shrubs, broadleaf evergreen shrubs, and sedges.

Line 117-122: Perhaps move this information to the Introduction section, to better frame the high ground evaporation bias as a problem that is addressed in this paper.

Section 2.1.2: The process for estimating parameter values for the new shrub PFTs seems a little unclear. It sounds as if parameter values developed by Wu et al. (2016) were used as a starting point for parameters in the biogeochemical sub-module. Were parameters further updated based on values published in the literature and ensuring that Equations (7) and (8) are satisfied?

Table 2: Can the parameters be categorized as physics sub-module vs biogeochemical sub-module?

Line 173: What is the area of the study site?

Line 191: Are the eddy covariance measurements of turbulent $CO_2$ flux and energy fluxes collected year-round? E.g., in Figures 4, 5, and 6, the eddy covariance observations are not shown for the winter period. The Table 5 caption states that observations were only available during the growing season. Perhaps this can be clarified in the main text.

Additionally, what is the approximate area of the eddy covariance tower footprint? Is the tower footprint area heterogeneous in terms of vegetation communities and soil wetness/surface water features, and would footprint area dynamics potentially impact measured fluxes?

Line 241: Over what time period are the meteorological observations available? (E.g., 2004-2017?)

Table 5: Do the observations reported here refer to only the eddy covariance tower measurements? When it is stated that observations were only available during the growing season, does this refer to uncertainties with the forced diffusion chambers being measured during only one winter and having limited spatial coverage?

Lines 412-417: Are simulations of GPP, Rh, and Ra all sensitive to the interannual

variability in meteorological forcing? Which component fluxes drive the switch to annual net CO2 sinks during certain years?

Line 442: Even with the new B formulation, ground E at the time of snowmelt remained overestimated. Is this based on comparison with field measurements? Is it inferred based on the simulated ponding during and after the snowmelt period?

Line 464: CLASSIC simulated ET to be 50 +/- 22% T for the last week of July 2004-2017 using the shrub PFT simulation with the new B formulation. Does this represent an improvement relative to the original B formulation?

Line 498: Here, do the three methods refer to eddy covariance, chamber, and model simulations?

Line 551 (and Line 410): Is the annual CO2 loss of 17 g C m-2 yr-1 estimated from the combined EC and chamber estimate the NEP value that is not shown in Table 5? Is this value not reported in Table 5 due to the uncertainties related to combining the two observation datasets?

Lines 565-572: Regarding the finding that growing season net CO2 uptake is more sensitive to environmental conditions in the grass PFT, are there differences in the amplitude of the CO2 flux seasonal cycle among the shrub, grass, and tree PFTs? E.g., among the different PFT simulations, does the timing of peaks in Reco and GPP – and any potential mismatch in timing of the Reco and GPP peaks – potentially shed light on changes in amplification of the net CO2 seasonal cycle?

Technical corrections:

Line 30: Need opening single quote at beginning of 'greening' Line 66: 'kind' needs to be pluralized. Line 108: Unclear how 'habits' is being used in this context. Figure 4. Caption: extra parentheses after d Line 19 in Supplement: It's stated that the PFT-dependent parameter Äś ÌĚ is given in Table 4, but is this meant to be Table 2? Line 413: t-test p-value one-sided or two-sided Line 463: t-test p-values one-sided or twosided

---

## Referee Comment (RC2) · Anonymous Referee #2 · 7 Mar 2021

General comments This manuscript addresses a plant functional type which is central to the functioning of many sub-Arctic and Arctic systems, but which is often overlooked. Thus, the objectives of this work are important and highly relevant to efforts to improve our understanding of Arctic carbon balance. I find this manuscript well written and clear and the work high quality. The model modifications described are well justified, and the methodology is broadly sound and appropriate. While in places I feel the text could benefit from some extra reader-guidance to navigate the length and detail of the manuscript (e.g. more subheadings), or perhaps from some editing to make the

discussion and parts of the results more concise, I have no substantial concerns with regard to the quality or communication of the work.

Specific comments Methods – Measurements and data processing: Some extra sub-headings would be helpful here, e.g. to separate out EC set up, soil chamber set up and CLASSIC runs. Soil chambers – did these remain closed throughout the summer and winter? If so, how did you prevent $CO_2$ build up above ambient, chamber heating and other artifacts? How did you measure and account for any artifacts of taking repeated measurements in unvented chambers? Detrital pool: Does the lability of litter differ between different functional types?

Technical corrections/suggestions Abstract L1: Large mouthful for a first sentence! Maybe condense slightly to something like: The Arctic is warming more rapidly than other regions of the world, leading to ecosystem change including shifts in vegetation communities, permafrost degradation and alteration of tundra surface-atmosphere energy and carbon (C) fluxes, among others changes. L61 change ',' after tundra to '.' L63 ',' after diverse Table 2: Really useful table, but would it be too disruptive to have a brief description for each parameter either in a table column or in the legend? Not critical and I know its reader laziness, but it would be extra helpful!

---

## Author Comment (AC2) · 7 Apr 2021

Our replies to the Reviewer's comments are in the attached PDF-document.

Please also note the supplement to this comment:
https://bg.copernicus.org/preprints/bg-2020-458/bg-2020-458-AC2-supplement.pdf

---

## Author Response (AR1)

**Author reply**

We thank the two reviewers for their time and helpful comments. Below we answer their questions (our replies are in blue) and describe any modifications we have made.

**Anonymous Referee #1**

General comments: The manuscript describes modifications to the Canadian Land Surface Scheme Including biogeochemical Cycles (CLASSIC) model, and it evaluates simulations of surface-atmosphere carbon dioxide ($CO_2$), water, and energy exchange against observations from eddy covariance and carbon flux chambers for a dwarf-shrub tundra site in Canada's Southern Arctic ecozone. The study added new shrub plant functional types (PFTs) to the physical and biogeochemical sub-modules of the CLASSIC model. Additionally, the CLASSIC model was modified to correct for a high ground evaporation bias which affects simulated energy fluxes and water-stress impacts on plant productivity. Simulations of the CLASSIC model were run to estimate annual and seasonal $CO_2$ fluxes, and model estimates were compared for three parameterizations (shrub, tree, and grass PFTs) and evaluated against observations. The ground evaporation bias correction, implemented as a new formulation of the coefficient B (used in the calculation of specific humidity at the surface), improved agreement between simulated and observed values for several variables (e.g., soil moisture in the upper soil layers, soil temperature in surface and deeper soil layers, spring and summer latent heat flux (LE), and $CO_2$ fluxes). When simulations with the new shrub PFTs were compared against simulations with grass or tree PFTs, the shrub simulations had the best agreement with observations for variables such as leaf area index (LAI), soil and litter carbon pools, vegetation height, mean rooting depth, summer soil temperature, and late growing season soil moisture. Notably, the shrub PFT improved simulation of vegetation burial by snow, which impacts net energy fluxes, phenology, and the timing and magnitude of net $CO_2$ fluxes. Together, the evaporation bias correction and new shrub PFTs led to improved agreement with observations considering the magnitudes of both net and component $CO_2$ fluxes. Results demonstrated the importance of incorporating appropriate PFTs in process-based models, particularly in terms of capturing the physical and biogeochemical impacts of shrubs in tundra ecosystems.

The manuscript is well written and well organized, and the lack of tundra-specific PFTs in the CLASSIC model is clearly framed as a problem to be addressed. The presented figures and tables provide clear evidence that the modifications to the CLASSIC model (both the addition of shrub PFTs and the correction of the ground evaporation bias) led to improved estimates of carbon and energy fluxes at the dwarf-shrub tundra site. The manuscript also acknowledged continuing modeling challenges, including overestimated net carbon uptake in the early growing season, overestimated winter $CO_2$ loss, and overestimated evapotranspiration during snowmelt (even with the new B formulation), and it provides suggestions for future model modifications that can address these challenges.

We thank the reviewer for their positive comments and obvious care taken in their review.

The authors might want to consider elaborating further on any significant trends present in the observed or modeled data over the 2004-2017 study period. For example, it would be helpful to report whether there were significant positive trends in growing season vegetation productivity, potentially indicating increases in shrub growth over the simulation period. Additionally, were temporal trends present in annual respiration, soil temperature, active layer depth, growing season length, or other variables? If significant trends are present, do they provide further information about physical or biogeochemical processes occurring at the site over the study period? Overall, the manuscript presents important modifications to the CLASSIC model that help to improve understanding of the drivers of $CO_2$ flux dynamics in shrub-dominated tundra ecosystems.

There were increasing trends in our simulations. Annual NEP, GPP, $R_a$, LE, top layer soil temperatures, active layer depths and precipitation had statistically significant increasing trends (p-values < 0.05) over the 2004-2017 period for all three simulations. For the grass and tree simulations, LAI also increased significantly, while there was no significant trend in LAI for the shrub simulation. There was a significant positive trend in observed growing season (May 1 – September 30) air temperature (which was used as a driving variable) and a positive trend in growing season soil temperature at 60 cm. Annual observed soil temperatures at 5 and 25 cm depths also showed statistically significant increasing trends for 2004-2017. Simulations appeared to respond to these warming trends but observed growing season NEP, GPP and $R_e$ did not show significant trends. This is not surprising as the measured fluxes were highly variable and likely responded to short-term interannual variability in meteorological variables along with other physical and biogeochemical processes not entirely captured by the model. We did not expect vegetation change to influence observed or simulated trends over the relatively short time period of 14 years. We have opted to keep the focus of this study on the evaluation of the new shrub plant functional types but look forward to exploring this topic in subsequent studies using CLASSIC.

Specific comments:
Line 42: It might be worthwhile to mention that some emissions observed during the winter could be driven by the diffusion of stored CO2 that was produced during the non-winter period (although the magnitude of the contribution is not known, e.g., Natali et al. 2019).

Thanks for your suggestion! We've included the following sentence (p. 2, ll. 43-45):
"In addition to belowground microbial respiration during the winter months, diffusion of stored $CO_2$ produced during the non-winter period could have contributed, to an unknown extent, to the observed winter $CO_2$ emissions (Natali et al., 2019)."

Lines 101-103: It sounds as if respiration processes Ra and Rh are determined as a function of C pools within the biogeochemical sub-module (i.e., daily timestep). Are daily respiration rates also determined as functions of soil temperature or other inputs? (E.g., Lines 514-515). What determines the base soil respiration rate (e.g., Line 517)? Is Ra estimated as a function of daily carbon assimilation?

Do the model parameters involved with estimating respiration vary by PFT and/or soil layer? It would also be helpful to briefly describe how changes to soil carbon pools are calculated in the different soil layers.

The model version used in this study uses a bulk soil C pool, so soil C is not tracked per layer. Model parameters used in determining $R_a$ and $R_h$ do, however, vary by PFT and we have added these parameters to Table 2 (see below). To clarify how the different respiration components are determined in the model and their dependencies on temperature and moisture, for example, we have added the following section to the Supplementary Materials (p.2-3, ll. 23-59).

In CLASSIC, the two components of ecosystem respiration ($R_e$; mol $CO_2$ m$^{-2}$ s$^{-1}$), autotrophic respiration ($R_a$; mol $CO_2$ m$^{-2}$ s$^{-1}$) from the live vegetation components and heterotrophic respiration ($R_h$; mol $CO_2$ m$^{-2}$ s$^{-1}$) from the litter and soil C pools, are determined separately. Calculation of the different respiration components is described in detail by Melton and Arora (2016). Briefly, $R_a$ is calculated as the sum of maintenance ($R_m$; mol $CO_2$ m$^{-2}$ s$^{-1}$) and growth respiration ($R_g$; mol $CO_2$ m$^{-2}$ s$^{-1}$). For the leaves, $R_m$ is calculated on a half-hourly time step, same as photosynthesis, while the stem ($R_{mS}$) and root ($R_{mR}$) components are calculated on a daily time step (Melton and Arora, 2016). Leaf $R_m$ ($R_{mL}$) is a function of the maximum rate of carboxylation by the Rubisco enzyme ($V_{cmax}$) adjusted for temperature and soil moisture limitations ($V_m$), $Q_{10d,n}$, the temperature sensitivity during the day(d) and night(n), and the photosynthetically active radiation (PAR) absorbed throughout the canopy, which scales up $R_{mL}$ to the canopy level,

$$R_{mL} = \varsigma_L V_m f_{25}(Q_{10d,n}) f_{PAR}, \tag{S7}$$

with the leaf maintenance respiration coefficient $\varsigma_L$ (kg C (kg C)$^{-1}$ yr$^{-1}$). The root and stem components of $R_m$ depend on the stem and root base respiration rates at 15°C ($\varsigma_S$ and $\varsigma_R$; kg C (kg C)$^{-1}$ yr$^{-1}$), the stem or root C mass ($C_i$; kg C m$^{-2}$) and the live fraction of the stem or roots ($l_{v,i}$) and are calculated as

$$R_{m,i} = 2.64 \times 10^{-6} \varsigma_i l_{v,i} C_i f_{15}(Q_{10}), \ i = S, R, \tag{S8}$$

where $2.64 \times 10^{-6}$ converts from kg C m$^{-2}$ yr$^{-1}$ to mol $CO_2$ m$^{-2}$ s$^{-1}$. $Q_{10}$ is constrained between 1.5 and 4.0 and is determined by the temperature-dependent function

$$Q_{10} = 3.22 - 0.046(\frac{15.0 + T_{\{S,R\}}}{1.9}), \tag{S9}$$

where $T_{\{S,R\}}$ is the stem or root temperature (°C). Air temperature is used as $T_S$ and the root temperature $T_R$ is determined from soil temperatures and the fraction of roots in each layer (Melton and Arora, 2016). After accounting for $R_m$, $R_g$ is determined as a fraction (15 % for all PFTs) of gross canopy photosynthesis. Soil ($C_H$; kg C m$^{-2}$) and litter C ($C_D$; kg C m$^{-2}$), soil temperature and moisture as well as PFT-dependent base respiration rates at 15°C ($\varsigma_i$; kg C (kg C)$^{-1}$ yr$^{-1}$; Table 2) determine $R_h$ as

$$R_{h,i} = 2.64 \times 10^{-6} \varsigma_i C_i f_{15}(Q_{10}) f(\Psi)_i, \ i = D, H, \tag{S10}$$

where $\Psi$ (MPa) is the soil matric potential with $f(\Psi)$ limiting $R_h$ when soil moisture is low or high. The temperature-dependent $Q_{10}$ used to determine $R_h$ is calculated as

$$Q_{10} = 1.44 + 0.56 \tanh[0.075(46.0 - T_i)], \ i = D, H, \tag{S11}$$

using the litter or soil C pool temperature (°C), respectively. The mean soil temperature of the rooting zone taking into account the root fractions in the different layers is taken to be the soil C pool temperature, while the litter temperature is determined as $T_D = 0.7\, T_1 + 0.3\, T_R$, as leaves, stems and roots all contribute to the litter C pool, where $T_1$ is the temperature of the top soil layer. Turnover of stem ($D_S$; kg C m$^{-2}$ day$^{-1}$) and root ($D_R$; kg C m$^{-2}$ day$^{-1}$) biomass depend on their respective PFT-dependent turnover timescales ($\tau_S$ and $\tau_R$; yr; Table 2). The amounts of C added to the litter pool are calculated as

$$D_i = C_i[1 - exp(-\frac{1}{365\tau_i})], \ i = S, R. \tag{S12}$$

The fraction of $R_{h,D}$, determined by the PFT-dependent humification factor $\chi$ (Table 2), transferred from the litter to the soil C pool is

$$C_{D \rightarrow H} = \chi R_{h,D}. \tag{S13}$$

Similar to the root distribution, C is assumed to follow an exponential distribution within the soil column, but is not explicitly tracked per layer (Melton and Arora, 2016).

Table 2 has been modified to include the PFT-dependent parameters used in the calculation of the respiration components and a description of all the parameters found in the Table.

**Table 2.** CLASSIC's biogeochemical parameter values for the PFTs used in this study including the new PFTs (Sedge, Broadleaf Evergreen Shrub and Broadleaf Deciduous Cold Shrub). Equation numbers refer to those in this text or the Supplementary Materials. The parameters are: $V_{cmax}$: maximum carboxylation rate by the Rubisco enzyme, SLA: specific leaf area, $\tau_L$: leaf life span, $lr_{min}$: minimum root to shoot ratio affecting the allocation of C, $\bar{\iota}$: parameter determining the root profile and rooting depths, $\eta$ and $\kappa$: parameters determining the minimum stem and root biomass required to support the green leaf biomass, $A_{max}$: maximum vegetation age affecting the intrinsic mortality rate, $\varsigma_D$: litter base respiration rate at 15°C, $\varsigma_H$: soil C base respiration rate at 15°C, $\chi$: humification factor determining fraction of C transferred from the litter to the soil C pool, $\varsigma_S$: stem base respiration rate at 15°C, $\varsigma_R$: root base respiration rate at 15°C, $\varsigma_L$: leaf maintenance respiration coefficient, $\tau_S$: turnover timescale for the stem, $\tau_R$: turnover timescale for the roots.

| Parameter | Equation | Units | Sedge | Broadleaf Evergreen Shrub | Broadleaf Deciduous Cold Shrub | C3 grass | Needleleaf Evergreen Tree | Needleleaf Deciduous Tree |
|---|---|---|---|---|---|---|---|---|
| $V_{cmax}$ | 6 | $\mu$mol $CO_2$ m$^{-2}$ s$^{-1}$ | 40 | 60 | 60 | 55 | 42 | 47 |
| SLA | S2 | m$^2$ kg$^{-1}$ | 10 | 8 | 15 | _a | _a | _a |
| $\tau_L$ | S1 | years | 1 | 2 | 1 | 1 | 5 | 1 |
| $lr_{min}$ | 8 | dimensionless | 0.30 | 1.68 | 1.68 | 0.50 | 0.16 | 0.16 |
| $\bar{\iota}$ | S6 | dimensionless | 9.50 | 4.70 | 5.86 | 5.86 | 4.70 | 5.86 |
| $\eta$ | 7 | dimensionless | 3.0 | 6.0 | 6.0 | 3.0 | 10.0 | 30.8 |
| $\kappa$ | 7 | dimensionless | 1.2 | 1.2 | 1.2 | 1.2 | 1.6 | 1.6 |
| $A_{max}$ | S3 | years | N/A$^b$ | 100 | 100 | N/A$^b$ | 250 | 400 |
| $\varsigma_D$ | S10 | kg C (kg C)$^{-1}$ year$^{-1}$ | 0.5260 | 0.4453 | 0.5986 | 0.5260 | 0.4453 | 0.5986 |
| $\varsigma_H$ | S10 | kg C (kg C)$^{-1}$ year$^{-1}$ | 0.0125 | 0.0125 | 0.0125 | 0.0125 | 0.0260 | 0.0260 |
| $\chi$ | S13 | dimensionless | 0.42 | 0.42 | 0.42 | 0.42 | 0.42 | 0.42 |
| $\varsigma_S$ | S8 | kg C (kg C)$^{-1}$ year$^{-1}$ | 0.0 | 0.090 | 0.055 | 0.0 | 0.090 | 0.055 |
| $\varsigma_R$ | S8 | kg C (kg C)$^{-1}$ year$^{-1}$ | 0.100 | 0.500 | 0.285 | 0.100 | 0.500 | 0.285 |
| $\varsigma_L$ | S7 | kg C (kg C)$^{-1}$ year$^{-1}$ | 0.015 | 0.025 | 0.020 | 0.013 | 0.015 | 0.021 |
| $\tau_S$ | S12 | years | 0.0 | 65.0 | 75.0 | 0.0 | 86.3 | 86.3 |
| $\tau_R$ | S12 | years | 3.0 | 11.5 | 12.0 | 3.0 | 13.8 | 13.2 |

[a]Determined by the model from the leaf life span.
[b]In CLASSIC, $A_{max}$ is not defined for grasses and sedges, as the age related mortality is not applied to these PFTs.

Line 111: Perhaps mention why these particular PFTs were selected to be added to the biogeochemistry sub-module? I.e., cold broadleaf deciduous shrubs, broadleaf evergreen shrubs, and sedges.

The three PFTs added to CLASSIC, cold broadleaf deciduous shrubs, broadleaf evergreen shrubs and sedges, were selected to represent the broad categories of vascular vegetation most commonly found in Arctic tundra (e.g. Walker et al., 2005) or the understory of other northern high-latitude ecosystems such as the boreal forest.
We have now included this statement in Section 2.1.2 (p. 4, ll. 115-117).

Line 117-122: Perhaps move this information to the Introduction section, to better frame the high ground evaporation bias as a problem that is addressed in this paper.

We have added the following sentence in the Introduction (p. 3, ll. 76-78) to provide some background on the topic.
"For example, Sun and Verseghy (2019) found that soil *E* was overestimated for mid-latitude shrublands during wet periods in spring, which led to underestimation of evapotranspiration (ET) and photosynthesis in the summer."

Section 2.1.2: The process for estimating parameter values for the new shrub PFTs seems a little unclear. It sounds as if parameter values developed by Wu et al. (2016) were used as a starting point

for parameters in the biogeochemical sub-module. Were parameters further updated based on values published in the literature and ensuring that Equations (7) and (8) are satisfied?

As you mentioned and we described in ll. 113-114, we started out with the parameterizations from Wu et al. (2016) and adapted them as required for upland shrubs following the literature and measurements at DL1. For example, the parameter ī determining the root profile was modified so that simulated rooting depths represented observations at DL1 and other upland tundra sites, which have deeper roots than peatland shrubs parameterized by Wu et al. (2016), where the high water table prevents deeper root penetration. The allocation parameters η and κ in Equation 7 were determined using measurements of shrub tundra stem, root and leaf biomass from the literature satisfying Equation 7 (see ll. 165-168). The parameters η and κ are not modified to satisfy Equation 8, however, dynamically determined allocation fractions for the different PFTs, which depend on PFT-dependent base allocation fractions as well as the vegetation's light and water status (described in detail in Melton and Arora, 2016), have to fulfill the requirements in Equations 7 and 8. Thus, C is preferentially allocated to roots, if the root to shoot ratio is lower than the minimum ratio. Minimum root to shoot ratio parameter values were obtained from the literature (Qi et al. (2019) for shrubs).

Table 2: Can the parameters be categorized as physics sub-module vs biogeochemical sub-module?

All the parameters included in Table 2 are used in the biogeochemistry sub-module. To make this clearer, we have modified the Table caption (p. 7) to "CLASSIC's biogeochemical parameter values for the PFTs used in this study including the new PFTs (Sedge, Broadleaf Evergreen Shrub and Broadleaf Deciduous Cold Shrub). Equation numbers refer to those in this text or the Supplementary Materials.".

Line 173: What is the area of the study site?

We added the following sentence to Section 2.2.2 (p. 8, ll. 200-204) to address this question: "The EC instrumentation is mounted to a mast 4.1 m above the surface, where 90% of the total flux originates within 178 ± 21 m (± 1 standard deviation) from the flux tower determined using the flux footprint parameterization of Kljun et al. (2004). The tundra was well represented by the soil and vegetation characteristics described above for at least 400 m in all directions of the flux tower and thus there was adequate fetch to represent this tundra type."

Line 191: Are the eddy covariance measurements of turbulent CO2 flux and energy fluxes collected year-round? E.g., in Figures 4, 5, and 6, the eddy covariance observations are not shown for the winter period. The Table 5 caption states that observations were only available during the growing season. Perhaps this can be clarified in the main text.

Due to power limitations at the site, eddy covariance measurements are not collected year-round (only between March/April and October and with consistent data coverage from May through September). Only the meteorological, soil temperature and soil moisture measurements were made year-round due to their low power demands.

To make it clearer when measurements were available, we modified the first sentence in Section 2.2.2 (p. 8, ll. 197-198) to "Eddy covariance measurements of turbulent $CO_2$ flux and energy fluxes, latent (LE) and sensible (H) heat flux, have been made at DL1 during the growing season since 2004."

Additionally, what is the approximate area of the eddy covariance tower footprint? Is the tower footprint area heterogeneous in terms of vegetation communities and soil wetness/surface water features, and would footprint area dynamics potentially impact measured fluxes?

We added the following sentence to Section 2.2.2 (p. 8, ll. 200-204) to address this question:

"The EC instrumentation is mounted to a mast 4.1 m above the surface, where 90% of the total flux originates within 178 ± 21 m (± 1 standard deviation) from the flux tower determined using the flux footprint parameterization of Kljun et al. (2004). The tundra was well represented by the soil and vegetation characteristics described above for at least 400 m in all directions of the flux tower and thus there was adequate fetch to represent this tundra type."

Line 241: Over what time period are the meteorological observations available? (E.g., 2004-2017?)

Yes, the meteorological observations were available year-round for 2004 through 2017. We've edited the sentence in Section 2.2.2 (p. 10, ll. 257) to clarify this, "CLASSIC runs were forced using 30-min meteorological observations at DL1 from 2004-2017".

Table 5: Do the observations reported here refer to only the eddy covariance tower measurements? When it is stated that observations were only available during the growing season, does this refer to uncertainties with the forced diffusion chambers being measured during only one winter and having limited spatial coverage?

Yes, the observations reported in Table 5 only include the eddy covariance (EC) measurements. The chamber measurements were only available during one winter, have a much smaller footprint, contained little vegetation within the chambers, and thus, cannot represent the tundra in the same way as the EC measurements and model simulations. To make this clearer, we have added the following sentence to the table caption (p. 19), "Eddy covariance flux measurements were not available through the winter and are only reported for the growing season."

Lines 412-417: Are simulations of GPP, Rh, and Ra all sensitive to the interannual variability in meteorological forcing? Which component fluxes drive the switch to annual net CO2 sinks during certain years?

In years where the site switched to a $CO_2$ sink, GPP, $R_h$ and $R_a$ all increased, however, GPP increased more than the respiration components. The increase in annual GPP was more pronounced for the grass and tree simulations than the shrub simulation, and grasses showed the strongest photosynthetic response to interannual variability in meteorological variables.

We edited a sentence in the discussion (p. 27, ll. 592-593) to expand on this, "In our study, the grass PFT simulation of NEP was more sensitive to environmental conditions, primarily through variations in growing season net $CO_2$ uptake (Figure 7), when simulated growing season GPP increased more than $R_a$ and $R_h$."

Line 442: Even with the new B formulation, ground E at the time of snowmelt remained overestimated. Is this based on comparison with field measurements? Is it inferred based on the simulated ponding during and after the snowmelt period?

The statement that ground $E$ during snowmelt remained overestimated with the new β formulation is based on comparison with observed ET (see Figure 5). Simulated ponding of water on the surface occurred longer than observed at the site and is likely one of the reasons for this overestimation, but our conclusion that ground $E$ was overestimated is due to the fact that it was the main contributor to ET during this time and exceeded observed ET. We have edited line 466 (p. 23) to clarify this discussion:
"Even with the new β formulation, total ET and its main contributor, ground $E$, remained overestimated at the time of snowmelt, as demonstrated by simulated ET exceeding observed ET, likely due to a lack of infiltration into the porous surface soil."

Line 464: CLASSIC simulated ET to be 50 +/- 22% T for the last week of July 2004- 2017 using the shrub PFT simulation with the new B formulation. Does this represent an improvement relative to the original B formulation?

We added the following sentence (p. 24, ll. 490-492) to highlight the difference from the original β formulation results, "This was a large increase over the shrub simulation with the original β formulation, where $T$ was only 6 ± 5 % of ET, as shrub growth was greatly suppressed due to limiting soil moisture."

Line 498: Here, do the three methods refer to eddy covariance, chamber, and model simulations?
Yes, we have now specified this in the text (p. 25, ll. 525), "Of the three methods used in this study (EC, chamber, and model simulations), the rate of $CO_2$ emissions throughout the winter and shoulder seasons at DL1 was least with the chamber method and greatest in the model simulations."

Line 551 (and Line 410): Is the annual CO2 loss of 17 g C m-2 yr-1 estimated from the combined EC and chamber estimate the NEP value that is not shown in Table 5? Is this value not reported in Table 5 due to the uncertainties related to combining the two observation datasets?

Yes, the estimate of an annual $CO_2$ loss of 17 g C $m^{-2}$ $yr^{-1}$ was determined by combining the EC and chamber measurements at DL1 and has much less certainty than the growing season NEP determined using EC measurement only (Table 5) for the reasons outlined in section 3.4.

We have clarified the sources of data used to arrive at this estimate (p. 21, ll. 430-435), "Bearing in mind the caveats discussed above regarding combining chamber and EC data streams, these simulated results were similar to an estimated annual NEP of -17 g C $m^{-2}$ $yr^{-1}$ obtained using these two sets of flux observations at DL1. This estimate of annual NEP was calculated from the sum of EC-based NEP (12 ± 5 g C $m^{-2}$) for the 5-month growing season (Table 5), EC-based NEP (-19 ± 1 g C $m^{-2}$) for the 81 days EC flux data were available during the shoulder seasons and chamber-based NEP (-10 g C $m^{-2}$) for the 131 winter days EC fluxes were not available (Figure 6)."

Table 5 only includes EC flux measurements for the reasons noted above in a previous question above Table 5.

Lines 565-572: Regarding the finding that growing season net CO2 uptake is more sensitive to environmental conditions in the grass PFT, are there differences in the amplitude of the CO2 flux seasonal cycle among the shrub, grass, and tree PFTs? E.g., among the different PFT simulations, does the timing of peaks in Reco and GPP – and any potential mismatch in timing of the Reco and GPP peaks – potentially shed light on changes in amplification of the net CO2 seasonal cycle?

As can be seen in Figure 6, averaged over 2004-2017 the amplitude of NEP, GPP and $R_e$ differs between the shrub, grass and tree simulations. The grass PFT showed the largest amplitude in NEP and began to photosynthesize and peaked later in the year than the shrub and tree PFTs with the shrub PFT being the earliest. As noted in a response above, GPP increased more during the second averaging period (2010-2017) than $R_e$ while these differences in component fluxes were smaller for the tree and shrub simulations (Figure R.1). There was no significant change in the difference between the timing of peaks in $R_e$ and GPP for any of the simulations between the two time periods 2004-2009 and 2010-2017 as well as over the 2004-2017 time period.

Differences in PFT parameters determining allocation of C to stems, leaves and roots and its sensitivity to soil moisture and LAI as well as differences in rooting depths (the grass and tree simulations have deeper roots than the shrubs) likely play an important role in the PFTs sensitivity to meteorological conditions. We've added a short sentence to highlight this (p. 27, ll. 593-595), "A key

factor increasing the sensitivity of the grass PFT's GPP to environmental conditions may be the lack of stems, enabling grasses to accumulate leaf mass more quickly than other PFTs."

[Figure]

*Figure R1. Daily mean net ecosystem productivity (NEP), gross primary productivity (GPP) and ecosystem respiration ($R_e$) for the shrub, grass and tree plant functional type simulations averaged over 2004-2009 and 2010-2017, respectively. Shaded areas show the standard deviation of the daily mean.*

Technical corrections:

Line 30: Need opening single quote at beginning of 'greening'      Corrected
Line 66: 'kind' needs to be pluralized.          Corrected.
Line 108: Unclear how 'habits' is being used in this context. Replaced "habits" with "characteristics"
Figure 4. Caption: extra parentheses after d     Corrected.
Line 19 in Supplement: It's stated that the PFT dependent parameter is given in Table 4, but is this meant to be Table 2? Corrected
Line 413: t-test p-value one-sided or two-sided  We performed the two-tailed t-test and have now noted this in the manuscript.
Line 463: t-test p-values one-sided or two-sided   Also corrected to now note the use of the two-tailed t-test.

**Anonymous Referee #2**

General comments This manuscript addresses a plant functional type which is central to the functioning of many sub-Arctic and Arctic systems, but which is often overlooked. Thus, the objectives of this work are important and highly relevant to efforts to improve our understanding of Arctic carbon balance. I find this manuscript well written and

clear and the work high quality. The model modifications described are well justified, and the methodology is broadly sound and appropriate. While in places I feel the text could benefit from some extra reader-guidance to navigate the length and detail of the manuscript (e.g. more subheadings), or perhaps from some editing to make the discussion and parts of the results more concise, I have no substantial concerns with regard to the quality or communication of the work.

We thank the Reviewer for their positive comments.

Specific comments Methods – Measurements and data processing: Some extra sub-headings would be helpful here, e.g. to separate out EC set up, soil chamber set up and CLASSIC runs. Soil chambers – did these remain closed throughout the summer and winter? If so, how did you prevent CO2 build up above ambient, chamber heating and other artifacts? How did you measure and account for any artifacts of taking repeated measurements in unvented chambers?

As suggested, we have added sub-headings to Section 2.2.2 (Measurements and data processing).

The eosFD forced diffusion chambers are based on the dynamic steady-state (flow-through) chamber method. $CO_2$ continuously diffuses from the small chamber headspace to the atmosphere through a gas-permeable membrane. Thus $CO_2$ concentrations within the chamber are intermediate to the soil and atmospheric concentrations (Risk et al. 2011, eosFD User Manual). Although we did not monitor soil temperature at the chamber locations, snow pit observations at the chambers were very similar to those of the surrounding tundra.

Detrital pool: Does the lability of litter differ between different functional types?

In CLASSIC, litter decomposition varies between the plant functional types (PFTs), as the humification factor, determining the transfer of C from the litter to the soil C pool, and the base respiration rate for litter at 15ºC are PFT-dependent parameters. We have included the PFT-dependent respiration parameters in Table 2 (p. 7) and included the relevant equations in the Supplementary Materials (p. 2-3, ll. 23-59).

Technical corrections/suggestions
Abstract L1: Large mouthful for a first sentence!
Maybe condense slightly to something like: The Arctic is warming more rapidly than other regions of the world, leading to ecosystem change including shifts in vegetation communities, permafrost degradation and alteration of tundra surface-atmosphere en-ergy and carbon (C) fluxes, among others changes. We have shortened the sentence to "Climate change in the Arctic is leading to shifts in vegetation communities, permafrost degradation and alteration of tundra surface-atmosphere energy and carbon (C) fluxes, among other changes."
L61 change ',' after tundra to '.' Corrected
L63 ',' after diverse Corrected
Table 2: Really useful table, but would it be too disruptive to have a brief description for each parameter either in a table column or in the legend? Not critical and I know its reader laziness, but it would be extra helpful! We have now added a brief description of the parameters in the Table caption.

References:

eosFD Forced Diffusion Chamber and Software - User Manual, EOSENSE Inc. [online] Available from: https://s.campbellsci.com/documents/us/manuals/eosfd.pdf (Accessed 10 March 2021), n.d.

Kljun N, Calanca P, Rotach MW, Schmid HP. (2004) A Simple Parameterisation for Flux Footprint Predictions. Bound-Layer Meteorol. 112, 503–523.

Risk D, Nickerson N, Creelman C, McArthur G, Owens J. 2011. Forced Diffusion soil flux: A new technique for continuous monitoring of soil gas efflux. Agric For Meteorol., 151, 1622–1631.

Walker DA, Raynolds MK, Daniëls FJA, Einarsson E, Elvebakk A, Gould WA, et al. 2005. The Circumpolar Arctic vegetation map. J Veg Sci., 16, 267–282.